# The Self-Consistent Theory of Neural Network Moments

## Abstract

This paper establishes a rigorous mathematical foundation for the statistical behavior of neural network parameter and gradient moments through self-consistent equations. We prove that the logarithmic moments exhibit a universal asymptotic decomposition governed by extremal statistics. This framework is extended to construct a joint partition function that unifies parameter and gradient statistics, revealing a topological phase distinction between states of correlated and uncorrelated extrema. The theory provides exact microscopic guarantees for finite networks while capturing emergent scaling behavior in large-scale systems.

## 1 Introduction

The statistical properties of a neural network's weights and gradients fundamentally dictate its optimization and generalization capabilities Neyshabur et al. (2017). While many existing theoretical frameworks rely on infinite-width asymptotic limits, empirical observations of practical, finite-sized networks frequently reveal the dominance of heavy-tailed distributions and highly localized features in the learning dynamics Gürbüzbalaban et al. (2021). To bridge this gap, we introduce a deterministic framework, grounded in extremal statistics and large deviations theory, for analyzing the exact moment distributions of any finite neural network.

We first establish a set of self-consistent equations governing the individual evolution of parameter and gradient moments. By analyzing their high-order limits, we prove that the logarithmic moments follow a universal asymptotic form that explicitly decomposes the network's macroscopic statistics into contributions from its single largest value (the extremum), its multiplicity, and the bulk spectrum of all other values. While the derivation of these individual limits relies on the well-established mathematical properties of the log-sum-exp function, they serve as the necessary foundation for our primary objective: characterizing the coupled optimization physics of training.

Our core theoretical novelty emerges when we move beyond isolated sets to quantify the structural dependence between the network's state (parameters) and its learning signal (gradients). We construct a joint partition function and define a **coupling term** $C(k, l)$. The asymptotic behavior of this term reveals a mathematically rigorous phase distinction in learning dynamics. We identify an **ordered phase**, where maximum parameters structurally align with maximum gradients, and a **disordered phase**, where they decouple. Because parameters and gradients in neural networks are causally bound by backpropagation, we demonstrate that this mathematical boundedness condition ($m_\cap \geq 1$) provides a strictly computable diagnostic for optimization health. The coupling term offers a tractable, zero-order alternative to complex information-theoretic measures Tishby & Zaslavsky (2017); McAllester & Stratos (2020) or expensive second-order landscape computations.

We develop this framework to unify and interpret several macroscopic phenomena. We provide a topological explanation for catastrophic forgetting Kirkpatrick et al. (2017) as a phase reversal, analyze grokking Power et al. (2022) through the lens of continuous spectral alignment, and demonstrate how architectural components like Normalization and Residual Connections act as structural priors that actively facilitate the ordered phase.

## 1.1 Summary of Contributions

To explicitly delineate the boundaries of our theoretical framework, we note that the single-variable moment limits are natural consequences of classical extremal statistics. Our fundamental contribution lies in introducing the joint formulation to formally map these extreme statistical deviations to the physical mechanics and optimization geometry of deep learning. Specifically, our main contributions are:

- **The Joint Partition Function and Phase Transition:** We introduce a novel coupling formulation $C(k, l)$ that explicitly measures the interaction between a network's capacity and its learning signal. We prove a necessary and sufficient topological condition ($m_\cap \geq 1$) for its boundedness, mathematically defining the phase transition between aligned feature learning and decoupled disorder. This coupling is a specific instance of a broader scale-invariant operator $\mathcal{C}_\phi$ that isolates structural alignment from trivial scale drift.

- **Kinematic and Geometric Grounding:** We strictly ground the physical semantics of these phases in optimization mechanics. We prove that the ordered phase is a deterministic kinematic consequence of parameters acting as temporal integrals of stationary gradients. Furthermore, we reveal that the disordered phase perfectly diagnoses pathological Hessian geometry, where massive weights are trapped in zero-curvature flat regions, while microscopic weights endure near-infinite curvature.

- **A Unified Measurement Spectrum and Task-Adaptive Diagnostics:** We generalize the coupling operator to a continuum of scale-invariant lenses, ranging from macroscopic (rank-based CDF) to microscopic (high-order monomials). This reveals a fundamental topological duality: the optimal diagnostic probe is task-dependent—sparse "crystalline" tasks (e.g., algorithmic reasoning) require high-order extremal lenses, while dense "fluid" tasks (e.g., image classification) call for low-order or rank-based measures. Within the power-law subfamily, we derive a continuous contact quality index $Q_\alpha^{(\text{pow})}$ that quantifies alignment health in finite networks and provides an early warning of phase transitions such as catastrophic forgetting or grokking. This framework offers practitioners a principled basis for selecting and interpreting statistical probes during training.

## 2 Theoretical Framework

### 2.1 Parameter Moments

**Definition 2.1** (Absolute Parameter Moments). Let $\Theta = \{\theta_1, \theta_2, \ldots, \theta_n\}$ denote the complete parameter set of a neural network, where $n = |\Theta| < \infty$. The $k$-th order absolute moment is defined as:

$$M(k) := \frac{1}{n} \sum_{i=1}^{n} |\theta_i|^k, \quad k \geq 0 \tag{1}$$

with the convention that $0^0 = 1$ and $0^k = 0$ for $k > 0$.

**Remark.** This definition naturally handles zero-valued parameters: they contribute nothing to moments of order $k > 0$ and correspond to a Dirac mass at $\lambda = \infty$ in the spectral representation, which does not affect the Laplace transform for any finite $k$. The convention $0^0 = 1$ ensures proper normalization $M(0) = 1$.

### 2.2 Existence of Moment Exponents

**Theorem 2.2** (Existence and Explicit Value of Moment Exponents). *For any finite-parameter neural network, the limit:*

$$\beta := \lim_{k \to \infty} \frac{\log M(k)}{k} \tag{2}$$

*exists, is finite, and equals:*

$$\beta = \log\left(\max_{1 \leq i \leq n} |\theta_i|\right) = \sup_{k > 0} \frac{\log M(k)}{k} \tag{3}$$

*Proof.* The proof is provided in Appendix A.1. □

**Theorem 2.3** (Remainder Convergence). *The limit:*

$$R^* := \lim_{k \to \infty} \left( \log M(k) - \beta k \right) \tag{4}$$

*exists, is finite, and equals:*

$$R^* = \log \frac{m}{n} \tag{5}$$

*where $m$ denotes the multiplicity of parameters with maximal modulus.*

*Proof.* The proof follows directly from the derivation in Appendix A.1. □

### 2.3 Self-Consistent Equation Formulation

The exact asymptotic decomposition:

$$\log M(k) = \beta k + R^* + \Delta(k), \quad k \to \infty \tag{6}$$

can be refined through spectral analysis. Define the decay rates $\lambda_i := \log(\theta_{\max}/|\theta_i|) > 0$ for $|\theta_i| < \theta_{\max}$. Then:

$$\Delta(k) = \log \left[ 1 + \sum_{j=1}^{n-m} w_j e^{-\lambda_j k} \right] \tag{7}$$

In the thermodynamic limit $n \to \infty$, this converges to (see Appendix A.2 for a rigorous proof under weakened regularity conditions):

$$\Delta(k) \to \log \left[ 1 + \int_0^\infty \rho(\lambda) e^{-\lambda k} d\lambda \right] \tag{8}$$

where $\rho(\lambda)$ is the spectral density satisfying $\int_0^\infty \rho(\lambda) d\lambda = \frac{n-m}{n}$.

**Intuitive Interpretation.** The decomposition in Eq. (6) reveals a fundamental statistical competition within the network. The linear term $\beta k$ represents the contribution from the single "loudest" parameter (the extremum $\theta_{\max}$), akin to a signal dominating the noise. Conversely, the integral term in Eq. (8) aggregates the "background" contribution from the vast majority of non-extremal parameters. For large moments $k$, the statistics are dominated solely by the extremum, leading to a "winner-takes-all" regime where the fine details of the distribution fade away. As $k$ decreases, the background spectrum $\rho(\lambda)$ begins to contribute significantly, acting like a thermal bath in statistical mechanics. This implies that high-order moments effectively act as a "spectral filter," isolating the network's most singular features from the bulk distribution.

### 2.4 Experimental Validation of Moment Decomposition

#### 2.4.1 Phenomenological Model: Bi-Exponential Residual Structure

Our experiments consistently reveal that the residual term, $\Delta(k)$, exhibits a bi-exponential decay across diverse architectures. Based on this strong empirical observation, we propose a phenomenological model to explicitly capture the leading-order behavior of this residual term. The goal is not to derive this form from first principles, but to construct a minimal model that effectively describes the observed data.

We propose the functional form:

$$\Delta(k) \approx \log \left[ 1 + A_1 e^{-\lambda_1 k} + A_2 e^{-\lambda_2 k} \right] \tag{9}$$

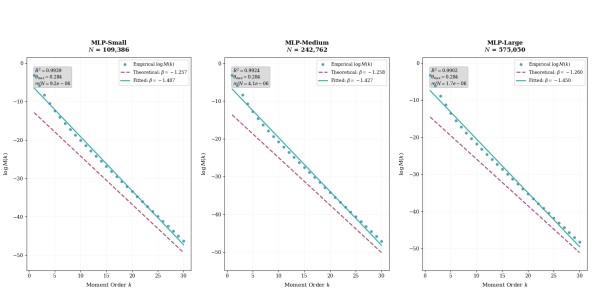

(a) Moment decomposition for MLP architectures.

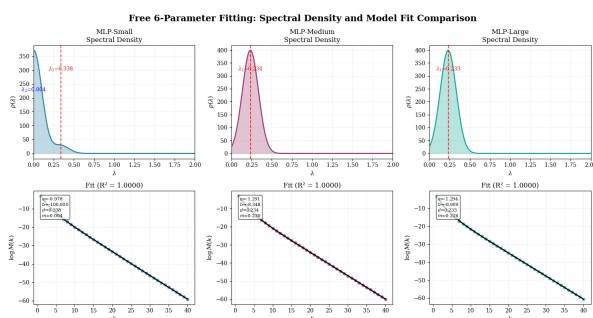

(b) Bi-exponential fitting of residual terms.

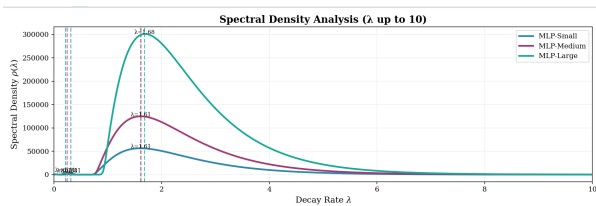

(c) Spectral density reconstruction from moments.

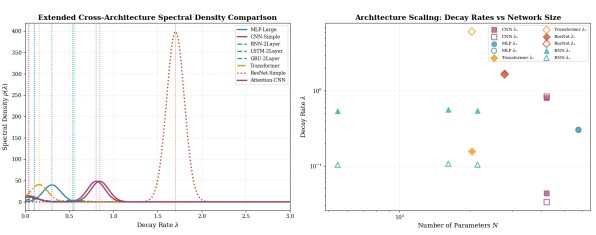

(d) Architecture-dependent spectral characteristics.

This form is motivated by its direct interpretation within our framework: it corresponds to a spectral density $\rho(\lambda)$ whose dominant features can be approximated by two discrete modes. This suggests that the vast number of non-extremal parameters tend to organize into distinct statistical ensembles, each with a characteristic decay rate, $\lambda_1$ and $\lambda_2$.

**Linearization and Spectral Interpretation** For large $k$, where the exponential terms are small, the logarithmic function can be linearly approximated via a first-order Taylor expansion, $\log(1 + x) \approx x$. This is not just a mathematical convenience; it provides a powerful tool for spectral interpretation. Applying this linearization to our model yields:

$$\log M(k) \approx \beta k + R^* + A_1 e^{-\lambda_1 k} + A_2 e^{-\lambda_2 k} \tag{10}$$

This approximation is empirically justified, as the condition $A_1 e^{-\lambda_1 k} + A_2 e^{-\lambda_2 k} \ll 1$ is validated by our moment analysis for sufficiently large $k$. The crucial insight here is that the linearized, empirically-fitted model directly reveals the structure of the underlying effective spectral density:

$$\rho_{eff}(\lambda) \approx A_1 \delta(\lambda - \lambda_1) + A_2 \delta(\lambda - \lambda_2) \tag{11}$$

This effective density, composed of two Dirac delta functions, should be understood as a simplified representation—a "two-peak" approximation—of what is likely a complex, continuous background spectrum. It successfully captures the dominant decay modes that govern the residual term's behavior at large $k$.

## 3 Gradient Moments and Statistical Isomorphism

### 3.1 Gradient Moment Theory

**Definition 3.1** (Absolute Gradient Moments). Let $\mathcal{G} = \{g_1, g_2, \ldots, g_n\}$ denote the gradient set corresponding to parameters. The $l$-th order gradient absolute moment is:

$$G(l) := \frac{1}{n} \sum_{i=1}^{n} |g_i|^l, \quad l \geq 0 \tag{12}$$

**Theorem 3.2** (Existence and Explicit Value of Gradient Moment Exponents). *For any finite-parameter neural network under gradient-based training, the limit:*

$$\beta_g := \lim_{l \to \infty} \frac{\log G(l)}{l} \tag{13}$$

*exists, is finite, and equals:*

$$\beta_g = \log\left(\max_{1 \le i \le n} |g_i|\right) = \sup_{l > 0} \frac{\log G(l)}{l} \tag{14}$$

*Proof.* The proof is analogous to that of Theorem 2.2 and is provided in Appendix A.3. □

**Remark on Applicability.** The gradient moment decomposition assumes regularity conditions (non-vanishing spectral gap and log-integrability) that typically hold in quasi-static training phases. Transient violations may occur during early training or in architectures with strong symmetries; see Appendix C for a complete characterization of these non-standard cases and their diagnostic value.

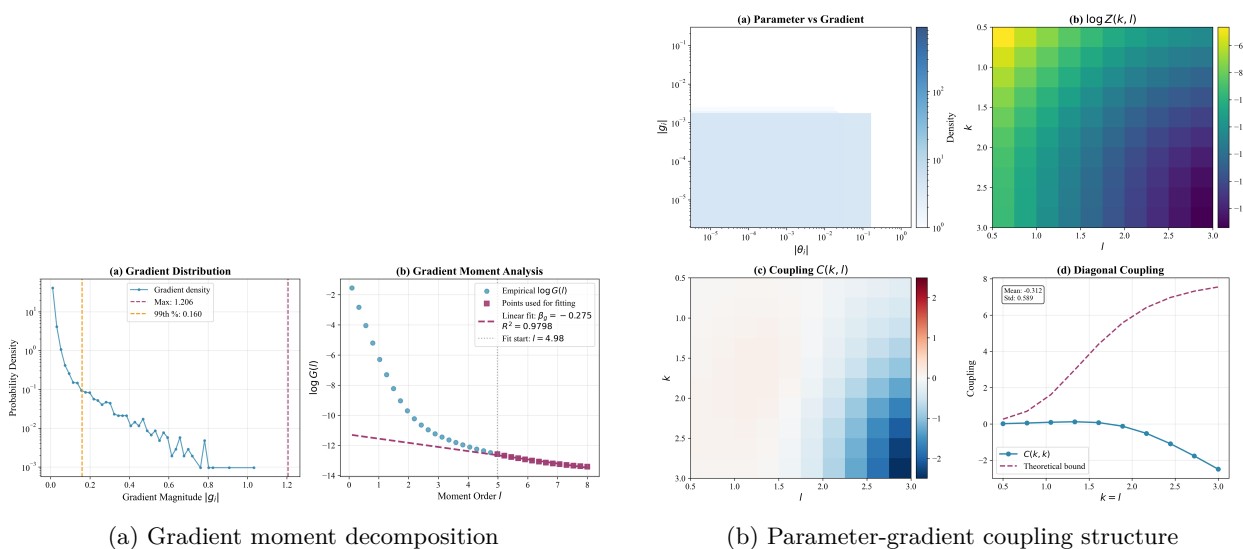

(a) Gradient moment decomposition      (b) Parameter-gradient coupling structure

Figure 2: Gradient moment statistics and coupling behavior. Deviations from predicted asymptotics may indicate regularity violations; see Appendix C.

## 4 Joint Partition Function: Theory of Bounded Coupling

### 4.1 Joint Moments and Decomposition

**Definition 4.1** (Joint Moments). For orders $k, l \ge 0$, the joint moment is defined as:

$$Z(k, l) := \frac{1}{n} \sum_{i=1}^{n} |\theta_i|^k |g_i|^l. \tag{15}$$

**Lemma 4.2** (Exact Decomposition of Logarithmic Joint Moments). *Let $\mathcal{C}(k, l) := \log Z(k, l) - \log M(k) - \log G(l)$ be the pure coupling term. Then:*

$$\log Z(k, l) = \log M(k) + \log G(l) + \mathcal{C}(k, l). \tag{16}$$

*This decomposition separates the joint statistics into marginal contributions and a term that captures their interaction.*

**The Generalized Coupling Operator and Scale Invariance.** The coupling term $\mathcal{C}(k,l)$ is a specific instance of a more general, scale-invariant operator that isolates structural alignment from trivial magnitude drift. Let $\phi : \mathbb{R}_+ \to \mathbb{R}_+$ be any monotonically increasing measurement function. We define the *generalized structural coupling* as

$$\mathcal{C}_\phi(\theta, g) := \log \frac{\mathbb{E}[\phi(|\theta|)\,\phi(|g|)]}{\mathbb{E}[\phi(|\theta|)]\,\mathbb{E}[\phi(|g|)]}. \tag{17}$$

During training, parameter and gradient norms often change due to learning rate schedules, normalization, the absence of weight decay, or intrinsic **weight symmetries (e.g., positive rescaling invariances in ReLU networks)**. A robust diagnostic must be insensitive to such trivial scale drift: for any $c_1, c_2 > 0$, we require $\mathcal{C}_\phi(c_1\theta, c_2 g) = \mathcal{C}_\phi(\theta, g)$.

This scale-invariance condition strongly constrains the admissible functions $\phi$. The only families satisfying it are:

- **Homogeneous functions** (e.g., monomials $\phi(x) = x^k$): the factors $c_1^k, c_2^k$ cancel between numerator and denominator.

- **Ordinal functions** (e.g., the empirical CDF $\phi(x) = \mathrm{CDF}(x)$): because $\mathrm{CDF}(cx) = \mathrm{CDF}(x)$, they are automatically scale-invariant.

All other functions (e.g., exponentials) would fail to cancel the scaling constants, entangling phase transitions with trivial weight inflation. Thus, the monomial family and the CDF constitute two fundamental, extreme points of a *generalized measurement spectrum*: the microscopic "extremal" lens ($x^k$ with large $k$) and the macroscopic "distributed" lens (rank/CDF). Intermediate behaviors are obtained by varying $k$ continuously between these extremes.

**Physical Semantics of the Phases: Why Neural Networks?** The coupling term $\mathcal{C}(k,l)$ serves as a rigorous diagnostic of structural alignment. A critical question arises: since the joint partition function mathematically applies to any arbitrary pair of arrays, why does the condition $m_\cap \geq 1$ specifically characterize "feature learning" in neural networks? The answer lies in the causal binding between parameters (representing the network's representational capacity) and gradients (representing the optimization driving force) via backpropagation.

When the system is in the **ordered phase** (bounded $\mathcal{C}$), the maximal learning signals ($|g|_{\max}$) are actively engaging with the network's most dominant features ($|\theta|_{\max}$). This structural alignment implies that the optimization energy is being efficiently utilized to refine the principal capacity of the network—the mathematical hallmark of healthy feature learning.

Conversely, a highly negative, divergent $\mathcal{C}(k,l)$ (**disordered phase**) implies a profound structural contradiction. The largest parameters (dominant features) receive vanishingly small gradients, becoming "frozen" and acting as wasted capacity. Concurrently, the maximum gradients (the most intense optimization updates) are applied to microscopic weights. The network expends its driving force violently oscillating background noise rather than refining principal features. Thus, the mathematical divergence of $\mathcal{C}(k,l)$ rigorously diagnoses the breakdown of the feature learning mechanism and the onset of noisy memorization.

In the remainder of this section we focus on the monomial subfamily $\phi(x) = x^k$, for which we will derive sharp asymptotic results (Theorems 4.6–4.8). The complementary macroscopic lens (rank/CDF) will be deployed together with the monomial family in Section 6 to demonstrate the full power of the generalized measurement spectrum and its task-dependent optimality.

## 4.2 Global Upper Bound on Coupling Term

**Theorem 4.3** (Cauchy-Schwarz Upper Bound)**.** *For any finite-parameter network, the coupling term satisfies:*

$$\mathcal{C}(k,l) \leq A(k) + B(l) \tag{18}$$

*where*

$$A(k) := \frac{1}{2} \log M(2k) - \log M(k), \tag{19}$$

$$B(l) := \frac{1}{2} \log G(2l) - \log G(l). \tag{20}$$

*Proof.* See Appendix A.4. □

**Corollary 4.4** (Boundedness of Coupling). *The coupling term is globally bounded from above:*

$$\mathcal{C}(k,l) \leq C_{\max} < \infty, \quad \forall k, l \geq 0. \tag{21}$$

*Proof.* See Appendix A.4. □

### 4.3 Non-existence of Universal Lower Bound

**Theorem 4.5** (Absence of Universal Lower Bound). *For any constant $C_{\min} \in \mathbb{R}$ and any network size $n \geq 2$, there exists a parameter-gradient configuration $(\Theta, \mathcal{G})$ such that $\mathcal{C}(k,l) < C_{\min}$ for some $k, l \geq 0$.*

*Proof.* See Appendix A.4 for a constructive proof. □

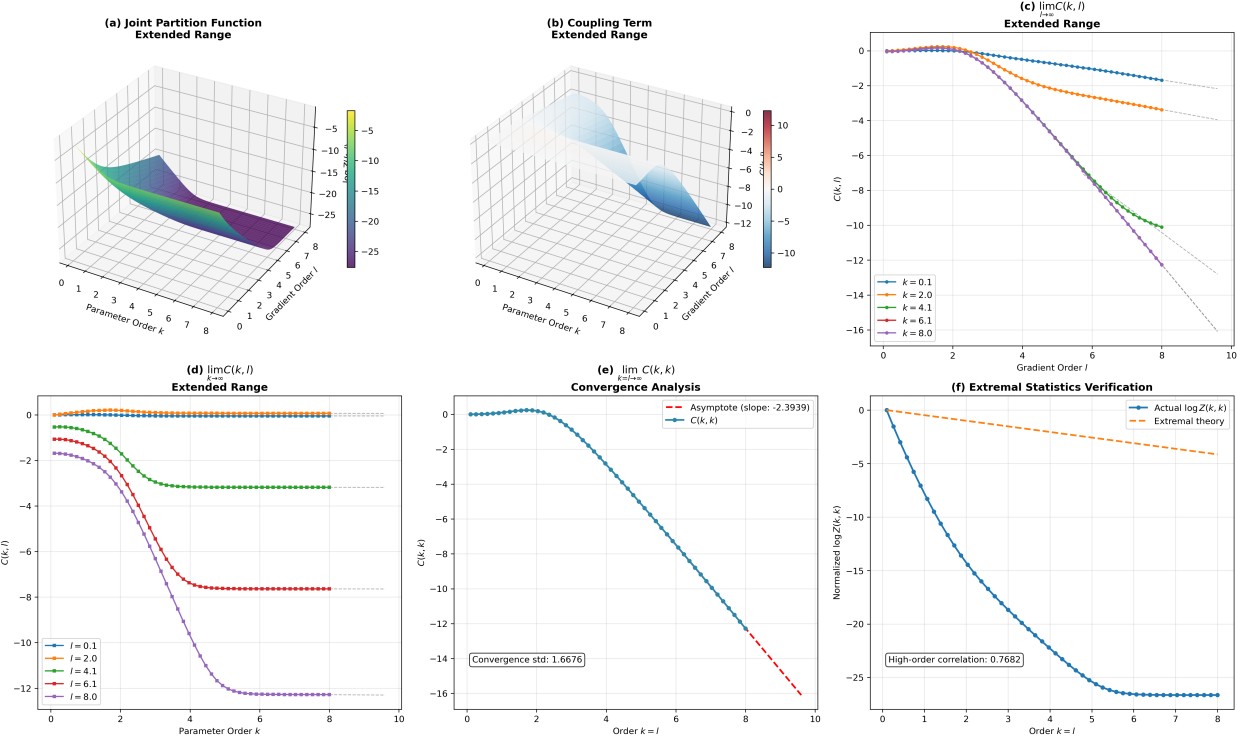

Figure 3: **Anisotropic Divergence in the Disordered Phase ($m_\cap = 0$).** The figure confirms that divergence in the disordered phase is directional (anisotropic). **(a, b)** The 3D surface of the coupling term $\mathcal{C}(k,l)$ visually confirms this, appearing bounded along the axes but plunging towards negative infinity along the diagonal. **(c, d)** Axial boundedness is confirmed by the unilateral limits, which converge to finite constants for any fixed order, despite showing transient downward trends. **(e)** In sharp contrast, the diagonal limit $\mathcal{C}(k,k)$ is clearly unbounded and follows a negative linear asymptote. **(f)** Throughout, the data respects a theoretical upper bound, confirming the model's mathematical consistency.

### 4.4 Asymptotic Analysis of Coupling

**Theorem 4.6** (Fixed Order Asymptotics). *For unilateral limits, we have:*

*(i) For any fixed $k \geq 0$:*

$$\lim_{l \to \infty} \mathcal{C}(k, l) = \log\left( \frac{1}{m_g} \sum_{i \in I_{\max}} |\theta_i|^k \right) - \log M(k).$$

*(ii) For any fixed $l \geq 0$:*

$$\lim_{k \to \infty} \mathcal{C}(k, l) = \log\left( \frac{1}{m} \sum_{i \in J_{\max}} |g_i|^l \right) - \log G(l).$$

*Where $J_{\max} = \arg\max_i |\theta_i|$ and $I_{\max} = \arg\max_i |g_i|$.*

*Proof.* The proof follows from applying the logic of Theorem 2.2 to the definition of $\mathcal{C}(k, l)$ in the specified limits. □

**Theorem 4.7** (Diagonal Asymptotics). *Let $k, l \to \infty$ with $l/k \to \alpha \in (0, \infty)$. The asymptotic behavior of the coupling term is determined by the intersection of the extremal sets, $m_\cap = |J_{\max} \cap I_{\max}|$.*

*(i) **Correlated Extrema** ($m_\cap > 0$): If the extremal sets intersect, the coupling term converges to a universal constant independent of $\alpha$:*

$$\lim_{\substack{k,l \to \infty \\ l/k \to \alpha}} \mathcal{C}(k, l) = \log\left( \frac{n \cdot m_\cap}{m \cdot m_g} \right)$$

*(ii) **Disjoint Extrema** ($m_\cap = 0$): If the extremal sets are disjoint, the coupling term diverges to negative infinity:*

$$\lim_{\substack{k,l \to \infty \\ l/k \to \alpha}} \mathcal{C}(k, l) = -\infty$$

*Proof.* See Appendix A.5. □

**Theorem 4.8** (Boundedness Condition and Lower Bound). *Let $\Theta = \{\theta_i\}_{i=1}^n$ and $\mathcal{G} = \{g_i\}_{i=1}^n$ be finite, non-zero parameter-gradient sets. Define the extremal sets $J_{\max} = \arg\max_i |\theta_i|$ and $I_{\max} = \arg\max_i |g_i|$. Let this system have **non-degenerate spectral gaps**, meaning the maximum values are strictly greater than all others:*

$$\theta_{\max} > \sup_{j \notin J_{\max}} |\theta_j| \quad and \quad g_{\max} > \sup_{i \notin I_{\max}} |g_i|. \tag{22}$$

*Under this condition, the coupling term*

$$\mathcal{C}(k, l) = \log Z(k, l) - \log M(k) - \log G(l) \tag{23}$$

*admits a finite lower bound for all $k, l \geq 0$ if and only if the extremal sets intersect, i.e.,*

$$\boxed{m_\cap := |J_{\max} \cap I_{\max}| \geq 1.} \tag{24}$$

*When this condition holds, a valid lower bound is given by $\log(m_\cap/n)$.*

*Proof.* See Appendix A.5. □

### 4.5 Joint Self-Consistent Equation Formulation

This necessary and sufficient condition allows for a complete spectral decomposition of the coupling term.

**Theorem 4.9** (Spectral Representation of Coupling Term). *Let the spectral decay rates be $\lambda_i = \log(\theta_{\max}/|\theta_i|)$ and $\mu_i = \log(g_{\max}/|g_i|)$. The coupling term $\mathcal{C}(k,l)$ admits the exact spectral decomposition:*

$$
\mathcal{C}(k,l) = \underbrace{\log \frac{m_\cap n}{mm_g}}_{\text{Topological Constant}} + \underbrace{\log \left[ 1 + \frac{1}{m_\cap} \sum_{i \notin K_{\max}} e^{-\lambda_i k - \mu_i l} \right]}_{\text{Joint Spectral Correction}}
$$
$$
- \underbrace{\log \left[ 1 + \frac{1}{m} \sum_{i \notin J_{\max}} e^{-\lambda_i k} \right]}_{\text{Parameter Spectral Correction}} - \underbrace{\log \left[ 1 + \frac{1}{m_g} \sum_{i \notin I_{\max}} e^{-\mu_i l} \right]}_{\text{Gradient Spectral Correction}}.
$$

(25)

*where $K_{\max} = J_{\max} \cap I_{\max}$. This form holds when $m_\cap \geq 1$.*

### 4.6 The Kinematic Origin of Alignment: Parameters as Temporal Integrals

To fundamentally understand why a healthy neural network deterministically gravitates toward the ordered phase ($m_\cap \geq 1$), we must shift from a static statistical view to a kinematic one. By the fundamental theorem of continuous-time gradient descent (gradient flow), a parameter at time $T$ is fundamentally the discrete time-integral of its historical gradients:

$$
\theta_i(T) = \theta_i(0) - \int_0^T \eta(t) g_i(t) dt
$$

(26)

where $\eta(t)$ is the learning rate. This temporal perspective allows us to formally ground the phase transition in optimization dynamics.

**Proposition 4.10** (Kinematic Alignment). *Assume the network undergoes feature learning in a stationary loss landscape, where the local gradient can be decomposed into a constant coherent drift (signal) $\mu_i$ and a bounded zero-mean noise (fluctuation) $\epsilon_i(t)$, such that $g_i(t) = \mu_i + \epsilon_i(t)$. If there exists a unique dominant feature index $i^*$ with maximum temporal coherence $|\mu_{i^*}| > |\mu_j|$ for all $j \neq i^*$, then for a sufficiently large training time $T$, the system is deterministically guaranteed to be in the ordered phase ($m_\cap \geq 1$).*

*Proof.* By integrating the gradient decomposition over time, the parameter magnitude at time $T$ is:

$$
|\theta_i(T)| = \left| \theta_i(0) - \int_0^T \eta(t) \mu_i dt - \int_0^T \eta(t) \epsilon_i(t) dt \right|
$$

Assuming a constant learning rate $\eta$ for simplicity, as $T \to \infty$, the parameter magnitude is strictly dominated by the linear drift term, since the integral of the zero-mean noise scales sub-linearly (analogous to a Wiener process, $\mathcal{O}(\sqrt{T})$):

$$
|\theta_i(T)| = \eta |\mu_i| T + \mathcal{O}(\sqrt{T}) + \mathcal{O}(1)
$$

Thus, as time accumulates, the index that maximizes the parameter magnitude is strictly determined by the maximum persistent drift: $J_{\max}(T) = \arg\max_i |\mu_i| = \{i^*\}$.

Simultaneously, the expected magnitude of the instantaneous learning signal (the current gradient) is also governed by its stationary drift component, as the expectation of the noise is zero: $\mathbb{E}[|g_i(T)|] \approx |\mu_i|$. This yields the expected gradient extremal set: $I_{\max}(T) = \arg\max_i |\mu_i| = \{i^*\}$.

Since both the historical integral maximum and the current derivative maximum are dictated by the same coherent signal, $J_{\max}(T) = I_{\max}(T) = \{i^*\}$. The intersection is non-empty ($m_\cap \geq 1$), which deterministically bounds the coupling term $\mathcal{C}(k,l)$. $\square$

This kinematic perspective elegantly previews the topological phase reversals observed in phenomena like **catastrophic forgetting** (detailed in Section 5.3). When a learning task shifts abruptly, the new instantaneous gradients $g(T)$ decouple entirely from the massive historical integral $\theta(T)$ established by the previous task. The coherent signal $\mu_i$ is shattered, causing a physical rupture between the network's capacity and its active gradients, which immediately drives $m_\cap \to 0$ and triggers the mathematical divergence of our joint partition function.

**Remark on Adaptive Optimizers (Adam/Momentum).** While Proposition 4.10 is derived under standard Gradient Descent, the kinematic logic fundamentally extends to modern adaptive optimizers like Adam. In Adam, the raw gradient $g_i(t)$ is replaced by the update vector $\hat{u}_i(t) \propto m_i(t)/\sqrt{v_i(t)}$, where $m_i$ and $v_i$ are exponential moving averages. Rather than breaking our framework, adaptive optimizers actually *accelerate and stabilize* the topological phase transition. If a parameter possesses a coherent drift $\mu_i$ (signal) amidst high-frequency batch noise $\sigma_i^2$, Adam acts as a temporal low-pass filter. It suppresses the noise variance and amplifies the coherent signal. By integrating this variance-normalized signal, the dominant capacity (maximal $\theta_i$) is even more strictly locked to the most statistically reliable features, thereby reinforcing the ordered phase ($m_\cap \geq 1$) against stochastic fluctuations.

# 5 Stability of Extremal Points Across Phases

The condition $m_\cap \geq 1$ from Theorem 4.8 is more than a mathematical curiosity; it marks a topological distinction between a "disordered" phase ($m_\cap = 0$, unbounded below) and an "ordered" phase ($m_\cap \geq 1$, bounded below). This distinction corresponds to a fundamental difference in the stability of extremal points under perturbations, such as those induced by training.

## 5.1 Unprotected Extremal Points in the Disordered Phase ($m_\cap = 0$)

In this phase, the set of largest parameters and the set of largest gradients are disjoint. This lack of alignment leads to fragility.

**Proposition 5.1** (Instability of Extremal Points). *When $m_\cap = 0$, for any $\epsilon > 0$ and any target value $C_{target} < 0$, there exists a small perturbation of the parameters and gradients that results in $\mathcal{C}(k,l) < C_{target}$ for some finite $k, l$.*

*Proof.* See Appendix A.6. □

**Interpretation**: In the disordered phase, the identities of the extremal parameters are not anchored to the learning signal (gradients). This can be thought of as a "liquid" state, where gradient descent can easily reassign which parameters become dominant. The lack of a lower bound on $\mathcal{C}(k,l)$ reflects that there is no "energy penalty" for decorrelating the parameter and gradient extrema.

## 5.2 Topologically Protected Extremal Points in the Ordered Phase ($m_\cap \geq 1$)

In this phase, at least one parameter is simultaneously extremal in both magnitude and gradient. This creates a form of topological protection.

**Proposition 5.2** (Rigidity of Extremal Points). *When $m_\cap \geq 1$, the property of having overlapping extrema is robust against small continuous perturbations that preserve the maximal parameter and gradient values. The identity of the parameters forming this extremal core cannot change without crossing a phase transition.*

*Proof.* See Appendix A.6. □

**Interpretation**: The ordered phase behaves like a "solid" state where at least one extremal parameter is locked to an extremal gradient. The finite lower bound on the coupling term acts as a confining potential, preventing the system from decorrelating its most important parameters from the learning signal. This stability is crucial for forming robust representations.

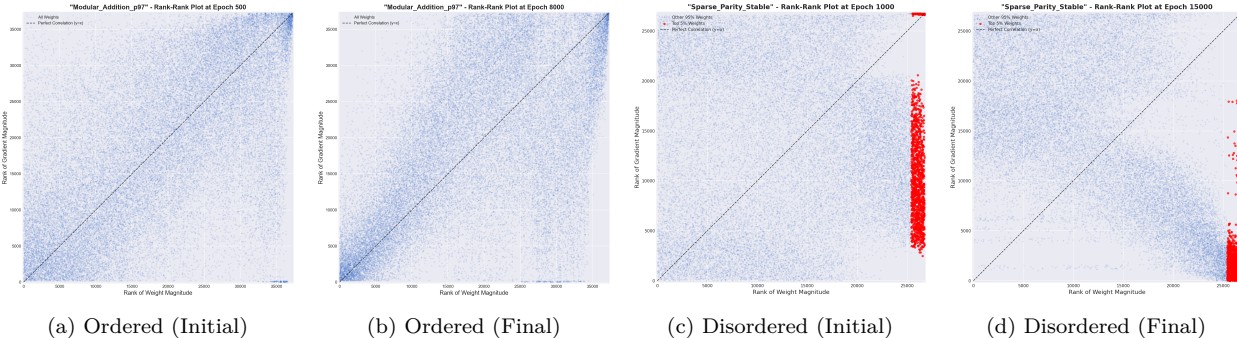

(a) Ordered (Initial)  (b) Ordered (Final)  (c) Disordered (Initial)  (d) Disordered (Final)

Figure 4: **Evolution of Topological Phases in Rank-Rank Space.** Plots of parameter magnitude rank (x-axis) vs. gradient magnitude rank (y-axis). **(a-b):** A successful run showing the evolution into an ordered phase, where ranks correlate along the diagonal, ensuring $m_\cap \geq 1$. **(c-d):** A failed run devolving into a disordered, anti-correlated state, where large weights get small gradients ($m_\cap = 0$).

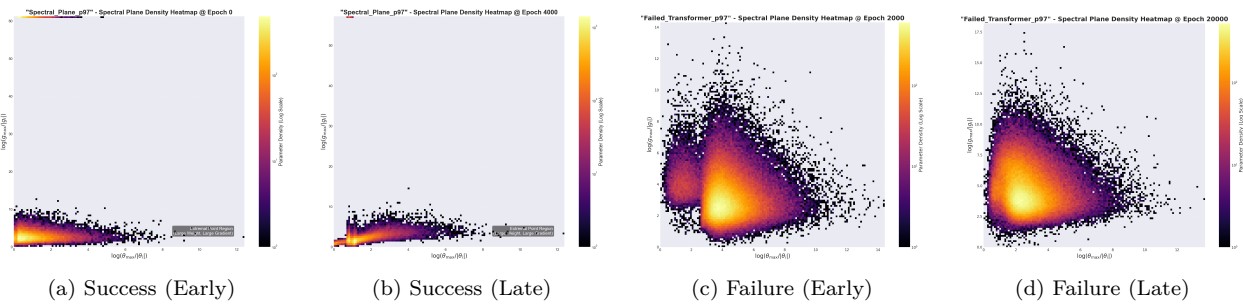

(a) Success (Early)  (b) Success (Late)  (c) Failure (Early)  (d) Failure (Late)

Figure 5: **Grokking vs. Failure in the Spectral Plane.** The plane plots log-decay from the max parameter (x-axis) and max gradient (y-axis). The origin (0,0) represents the ideal state of high-magnitude, high-gradient parameters. **Successful grokking (a-b)** is marked by density condensing at the origin over time. **Failure (c-d)** is characterized by density pathologically avoiding the origin, indicating large weights receive no learning signal.

### 5.3 Catastrophic Forgetting as Phase Reversal

The stability of the ordered phase ($m_\cap \geq 1$) is locally robust but globally fragile. A strong enough perturbation, such as training on a new, unrelated task, can shatter the alignment between parameters and gradients, inducing a phase transition from ordered to disordered ($m_\cap = 0$). This provides a topological explanation for catastrophic forgetting.

This stability is fundamentally linked to the **stability of the extremal sets** ($J_{\max}, I_{\max}$) against perturbations from the learning process. The degree of this stability is governed directly by the **spectral gaps** of the parameter and gradient distributions. A larger gap implies greater resilience, as a stronger perturbation is required to alter the membership of these extremal sets and risk a phase reversal. In the limit of maximum stability, where the spectral gap is maximized, the system approaches a state of **Neural Collapse**. This regime, where all class-relevant extremal features coalesce onto a maximally separated simplex structure, represents the most robust possible form of the ordered phase. It is, therefore, maximally resistant to catastrophic forgetting (a formal treatment of this connection is deferred to Appendix B).

To test this phase reversal hypothesis, we conducted a continual learning experiment on sequential MNIST (Task A: digits 0-4; Task B: digits 5-9). We compared a baseline with our theory-guided Elastic Weight Consolidation (EWC), which uses a penalty to elastically protect the extremal core stability of Task A.

The results in Table 1 confirm the hypothesis. The baseline model undergoes a complete phase reversal, losing its ordered structure for Task A. In contrast, our EWC approach acts as a "confining potential" that preserves the extremal set stability (and thus $m_\cap \geq 1$), demonstrating that catastrophic forgetting can be understood and mitigated as a controllable topological phase transition.

Table 1: Final Accuracy after Sequential Training on MNIST

| Strategy | Final Task A Accuracy | Final Task B Accuracy |
|---|---|---|
| Baseline (Fine-tuning) | 0.3069 | **0.9916** |
| Hard-Freeze Top-1000 | **0.9889** | 0.3364 |
| **EWC ($\lambda = 100$)** | **0.6698** | **0.9831** |

## 6 Controlled Divergence and Quasi-Ordered Phases in Practical Networks

The sharp phase transition between ordered ($m_\cap \geq 1$) and disordered ($m_\cap = 0$) phases is an idealization that emerges in the strict thermodynamic limit. In practical, finite-sized networks, the system often resides in an **intermediate state of approximate alignment**, where the largest parameters and gradients are not perfectly coincident but rather approach each other asymptotically. Our continuous framework naturally captures this realistic scenario through the geometry of the joint spectral support near the origin.

Rather than a discrete topological switch, transitions in finite networks are characterized by the **distance** of the spectral support from the origin and its contact order. This leads to a spectrum of critical behaviors ranging from **quasi-ordered** (logarithmic divergence) to **disordered** (linear divergence), providing a quantitative mathematical bridge between theoretical extremes and empirical training dynamics.

### 6.1 Approximate Alignment and Contact Stability

**Definition 6.1** (Spectral Gap and Contact Regime)**.** Consider a network with joint spectral measure $\nu$ supported on $F \subset \mathbb{R}_+^2$. We define:

- **Spectral Gap:** $d_0 := \text{dist}((0,0), F)$, measuring the minimal separation from perfect alignment.

- **Contact Regime:** If $d_0 = 0$ but $(0,0) \notin F$, the support *touches* the origin without containing it, defining a **quasi-ordered** phase.

When $d_0 > 0$, the system remains in a disordered phase with decoupled extrema. As training progresses, $d_0$ typically decreases, and the support may approach the origin with a characteristic **contact exponent** that quantifies the rate of alignment.

### 6.2 Critical Exponent and Asymptotic Divergence

The geometry of the support near the origin determines the asymptotic behavior of the coupling term. We characterize this geometry by the *contact function* and its scaling exponent.

**Definition 6.2** (Contact Exponent)**.** Assume the support near the origin satisfies a power-law scaling:

$$\inf\{\lambda + \mu : (\lambda, \mu) \in F, \ \lambda + \mu \geq r\} = Cr^\alpha + o(r^\alpha), \quad r \to 0^+$$

where $\alpha > 0$ is the **contact exponent**. Smaller $\alpha$ indicates a "flatter" approach to the origin, while $\alpha \to \infty$ corresponds to sharp, pointwise contact.

**Theorem 6.3** (Controlled Divergence in Quasi-Ordered Networks)**.** *Consider a network with approximate alignment characterized by spectral gap $d_0$ and contact exponent $\alpha$ (when $d_0 = 0$). The diagonal asymptotic behavior ($l = \alpha k$) of the coupling term is given by:*

$$\mathcal{C}(k,k) \sim \begin{cases} -\dfrac{2}{\alpha + 1} \log k + O(1), & d_0 = 0 \quad \textit{(quasi-ordered)} \\[2mm] -d_0 k + O(\log k), & d_0 > 0 \quad \textit{(disordered)} \end{cases}$$

The first case reveals that even without perfect alignment ($p_{00} = 0$), a *quasi-ordered* network exhibits only **logarithmic divergence**, which is far milder than the linear divergence of the fully disordered phase. This captures the **progressive alignment** observed during training, particularly in phenomena like grokking where the model slowly transitions from disorder to order.

### 6.3 From Theoretical Limits to Practical Measurement Spectra

The contact exponent allows us to define a theoretical continuous index for alignment quality:

$$Q_\alpha := \lim_{k \to \infty} \frac{-\mathcal{C}(k,k)}{\log k} = \frac{2}{\alpha + 1} \in [0, \infty]$$

where $Q_\alpha \approx 0$ indicates near-perfect alignment, and $Q_\alpha \to \infty$ indicates decoupled disorder.

**The Shift to Empirical Diagnostics.** While $Q_\alpha$ perfectly parameterizes the theoretical geometry of the approach to the origin, dynamically estimating a stable mathematical limit ($k \to \infty$) during the chaotic noise of Stochastic Gradient Descent is notoriously challenging. Instead of relying on this asymptotic proxy, our framework suggests a more direct and robust empirical approach: **monitoring the structural coupling $\mathcal{C}(k,k)$ itself at specific, finite observation scales.**

Rather than a single scalar $Q_\alpha$, evaluating $\mathcal{C}(k,k)$ at a sufficiently large order (e.g., $K = 20$), or shifting the basis entirely to the macroscopic Cumulative Distribution Function (CDF), yields a rich, observable **Measurement Spectrum**. The trajectory of these finite-order coupling terms acts as a highly sensitive "topological radar" during training.

**Practical Computability and Threshold Guidance.** A naive computation of high-order moments like $\sum |\theta_i|^{20}$ fundamentally triggers floating-point overflow. To ensure strict numerical stability for practitioners, we compute the logarithmic moments directly in the log-domain utilizing the Log-Sum-Exp (LSE) trick (detailed derivation is provided in **Appendix D**).

For empirical tracking, we recommend $K = 20$ as a standard microscopic lens. Based on our evaluations across various architectures, practitioners can observe the following transition:

- **Failure/Memorization:** $\mathcal{C}(20, 20) \ll 0$ (e.g., $-10$ to $-40$). The system is locked in the disordered phase.

- **Transition:** $\mathcal{C}(20, 20)$ exhibits a sustained, monotonic climbing trend, indicating continuous spectral alignment.

- **Generalization:** $\mathcal{C}(20, 20)$ breaks through zero and stabilizes at a positive value, confirming $m_\cap \geq 1$.

### 6.4 Practical Significance and Diagnostics

By directly tracking the finite-order microscopic coupling $\mathcal{C}(K, K)$ (e.g., $K = 20$), the theoretical transitions map strictly to observable training phenomena:

**Progressive Learning (Grokking).** During the "dark period" of grokking, macroscopic losses plateau, but the micro-structure is highly active. The system resides in a quasi-ordered state. Empirically, this is observed as the microscopic coupling $\mathcal{C}(20, 20)$ steadily climbing from deep negative divergence (disordered noise) towards zero, eventually spiking into a strong positive value the exact moment algebraic crystallization (order) occurs.

**Catastrophic Forgetting.** When a model switches tasks and catastrophic forgetting occurs, the new gradients instantly decouple from the historical parameter integrals. This topological rupture is immediately diagnosed by $\mathcal{C}(K, K)$ plunging back into severe negative divergence, providing an early, parameter-free warning of representational collapse.

**Architecture Design.** Architectures that promote higher intrinsic $\alpha$ values—such as weight sharing or skip connections—structurally prevent the spectrum from decaying into the linear divergence regime ($d_0 > 0$), facilitating a faster climb of $\mathcal{C}(K, K)$ toward the ordered phase.

## 6.5 Interaction with Modern Architectural Components

Our continuous transition framework provides a physical basis for understanding the success of ubiquitous deep learning components. We interpret these components as mechanisms that actively manipulate the spectral support to escape disorder and favor the ordered phase ($\mathcal{C}_\phi$ boundness).

**Normalization Layers (BatchNorm/LayerNorm).** Normalization techniques explicitly constrain the moments of the parameter distribution. In our framework, LayerNorm effectively imposes a hard constraint on the second moment $M(2) \approx 1$. This constrains the potential range of the extremal value $\theta_{\max}$ (Eq. 3), preventing the "runaway" of any single parameter. Crucially, this compression forces the system to maintain a compact spectral support, reducing the effective distance $d_0$ to the origin. By preventing spectral dispersion, normalization layers act as "confinement potentials" that stabilize the quasi-ordered phase and facilitate the transition to full alignment.

**Weight Decay as Spectral Filtering.** Standard weight decay ($L_2$ regularization) applies a penalty proportional to $\theta_i^2$. In terms of our spectral decomposition (Eq. 8), this acts as a "soft filter" that preferentially suppresses the heavy tail of the spectral density near $\lambda \approx 0$ (large weights). This effectively increases the parameter spectral gap $\Delta_\theta$. According to our stability analysis, a larger spectral gap enhances the rigidity of the extremal sets against perturbations, thereby increasing the robustness of the alignment against the noise of stochastic gradient descent.

**Residual Connections (ResNets).** Deep networks without residual connections often suffer from vanishing gradients, which in our framework corresponds to a degenerate gradient spectral gap ($\Delta_g \to \infty$). Residual connections create "gradient superhighways," ensuring that gradient magnitudes $|g_i|$ do not vanish exponentially with depth. This preservation of gradient magnitude scales is crucial for maintaining a well-defined $\beta_g$ (Eq. 13) across all layers, preventing the system from collapsing into the disordered phase and maintaining $m_\cap \geq 1$.

## 6.6 The Generalized Measurement Spectrum: Topological Duality in Practice

Building on the insight that finite-scale evaluations provide the most robust diagnostics, Section 4 established that the coupling operator $\mathcal{C}_\phi$ mathematically isolates topological alignment from trivial scale drift when constrained to the scale-invariant basis: the macroscopic Cumulative Distribution Function (Rank) and the microscopic monomials ($\phi(x) = x^p$). We now deploy this unified theoretical framework to diagnose real-world training dynamics. By continuously tuning the function $\phi$ from a concave, macroscopic lens (Rank, $p = 0.2$) to an extreme convex, microscopic lens ($p = 20.0$), we construct a **Generalized Measurement Spectrum**.

To demonstrate the universality and physical implications of this spectrum, we evaluate it across two fundamentally opposite learning paradigms: the sparse, algorithmic crystallization of Grokking (modular arithmetic), and the dense, distributed feature learning of CIFAR-10 (vision). The results, presented in Figure 6, reveal a profound topological duality.

**The Algorithmic Crystal (Grokking).** As shown in the top panel of Figure 6, algorithmic tasks require the network to learn a highly sparse set of exact logical circuits. Early in training, the network memorizes the dataset via dense, unstructured noise. Consequently, the macroscopic lenses (Rank) are easily saturated by this bulk noise, forming a premature structural plateau long before actual generalization occurs. However, as $p \to 20$, the extreme convex lens strictly filters out the bulk. In this thermodynamic limit, the metric $\mathcal{C}_{x^{20}}$ remains deeply frozen in the disordered phase for thousands of epochs, entirely ignoring the 100% training accuracy. It only exhibits a violent, critical phase transition exactly when the task-relevant, extremal algebraic circuits finally crystallize.

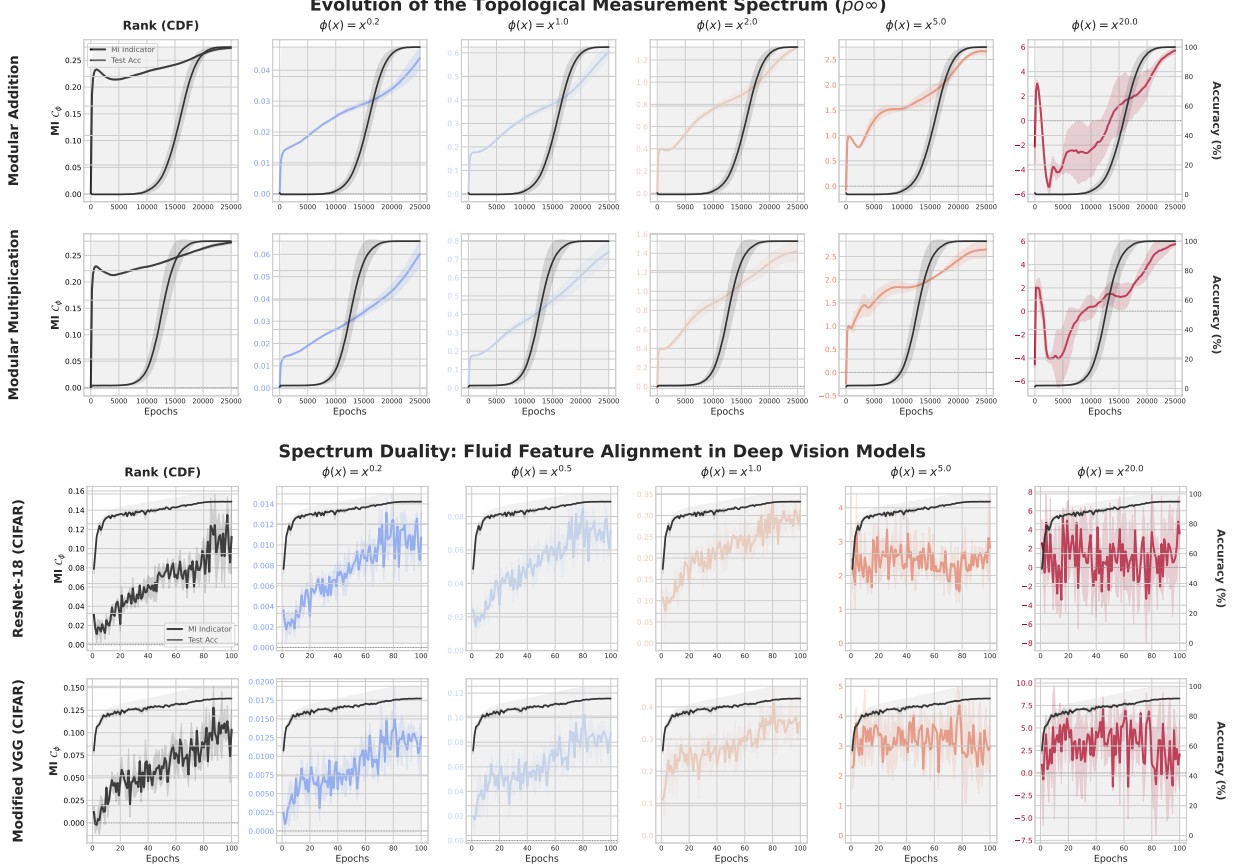

Figure 6: **Topological Duality across Learning Regimes.** We deploy the identical, unnormalized topological spectrum $\mathcal{C}_\phi$ on two contrasting tasks. **(Top) Grokking:** The macroscopic lenses (Rank, $p \leq 1$) suffer from premature saturation due to dense memorization noise. Conversely, the microscopic lens ($p = 20$) remains in deep negative divergence (disordered phase, conforming to Theorem 4.7) until it precisely spikes to predict the crystallization of sparse algebraic circuits. **(Bottom) CIFAR-10:** The duality perfectly inverts. The microscopic lens ($p = 20$) collapses into chaotic oscillation, blinded by local gradient noise at the $L_\infty$ boundary. Simultaneously, the macroscopic lenses (Rank, $p \leq 0.5$) elegantly and smoothly track the test accuracy, proving that vision representation is a distributed, fluid-like alignment.

**The Vision Fluid (CIFAR-10).** Strikingly, when we apply the identical mathematical spectrum to dense visual tasks (Figure 6, bottom panel), the topological narrative completely inverts. In the dense representation manifold of ResNet and VGG, generalization is not driven by a single "dictator" parameter, but by the collective statistical alignment of thousands of **distributed** micro-features. **Furthermore, these architectures possess extensive weight symmetries that inherently dilute single-coordinate extrema.** Consequently, the microscopic extremal lens ($p = 20$) completely degenerates. Because it relies exclusively on the $L_\infty$ norm, it is blinded by the transient, high-frequency batch noise of SGD, resulting in chaotic oscillations that fail to track any global learning progress. Conversely, the macroscopic lenses (Rank, $x^{0.5}$) elegantly emerge from the noise, producing a smooth, monotonic curve that perfectly shadows the test accuracy.

This duality conclusively validates our theoretical framework. It demonstrates that generalization cannot be universally captured by a single, static scalar metric. The valid topological indicator is not arbitrary; it strictly depends on the intrinsic physical dimensionality—whether crystalline (sparse) or fluid (dense)—of the underlying representation manifold. By merely swapping the scale-invariant function $\phi$, our joint partition function seamlessly bridges these two extremes.

We note that the distinct topological trajectories depicted in both learning regimes—the sharp extremal spike in Grokking and the smooth macroscopic emergence in CIFAR-10—are highly robust phenomena. To rigorously verify this, the evaluations in Figure 6 were conducted across 3 independent random seeds. The resulting trajectories consistently reproduce these exact functional geometries with negligible variance in the phase transition timing, confirming that our measurement spectrum captures intrinsic physical properties of the models rather than stochastic artifacts.

# 7 Discussion and Future Directions

The framework presented in this paper offers a deterministic lens through which to view the statistical mechanics of finite-sized neural networks. By analyzing the joint moments of parameters and gradients, we have uncovered a rich topological phase structure governed by structural alignment. While Section 4.6 grounds this alignment in optimization kinematics, fully unifying these macroscopic statistics with microscopic loss geometry remains an exciting frontier. Here, we delineate the theoretical implications of our unified spectrum and illuminate avenues for future research.

**The Geometric Origin: Taylor Expansion and Hessian Curvature.** While we proved the kinematic necessity of the ordered phase, its microscopic physical necessity can be rigorously understood through the local geometry of the loss landscape. Consider a Taylor expansion around a local minimum $\theta^*$, where the gradient is dominated by the Hessian $\mathcal{H}$: $g \approx \mathcal{H}(\theta - \theta^*)$. Under a locally diagonalized approximation, $g_i \approx \lambda_i \Delta\theta_i$, where $\lambda_i$ is the local curvature. Viewed through this geometric lens, the disordered phase ($m_\cap = 0$) implies a pathological spectral inversion: massive parameters (dominant features) correspond to near-zero gradients, implying $\lambda \approx 0$. They are trapped in flat, uninformative valleys, acting as "dead features." Conversely, microscopic parameters (background noise) endure the network's maximum gradients. For a tiny parameter to maintain a massive gradient, the underlying curvature must be extraordinarily high ($\lambda \to \infty$). Dynamically, to endure such massive updates yet remain microscopic, the parameter's trajectory must be zero-mean and variance-dominated—violently oscillating across a sharp ravine to fit high-frequency batch noise. Thus, $m_\cap = 0$ perfectly diagnoses a complete geometric collapse: structural capacity is wasted in flat regions, while optimization stress is entirely borne by hyper-sensitive noise. (*A formal mathematical translation of this geometric collapse is provided in* **Appendix E**).

**Open Question 1: Spectral Alignment and Generalization Bounds.** While we have grounded the phase transition in optimization kinematics, fully unifying these macroscopic statistics with exact generalization bounds (e.g., PAC-Bayes) remains an exciting open problem. We **hypothesize** that the pathological geometry of the disordered phase provides a structural link to sharpness. In classical generalization bounds, large weights incur a heavy complexity penalty. In the ordered phase, this penalty appears justified because the steepest constraints of the loss landscape ($\lambda_{\max}$) are correctly anchoring the network's dominant capacity. This corresponds to the network's eigenspectrum escaping the isotropic noise of initialization to form structured, informative spikes. In the disordered phase, the network accumulates a massive theoretical

capacity penalty (large frozen weights) without deriving representational utility from it, while simultaneously overfitting via high-curvature noise parameters. Consequently, **future work** will explore whether the contact quality index $Q_\alpha$ can formally serve as a computationally cheap, zero-order proxy for Hessian-based generalization capacity.

**Open Question 2: Model Scaling and the Emergence of LLM Outliers.**  A highly promising frontier for this extremal framework is the analysis of model scaling, particularly in modern Large Language Models (LLMs). Recent empirical studies (e.g., Dettmers et al., 2022) have revealed that as Transformer models scale up, they universally develop "massive outliers"—a tiny fraction of parameters and activations that grow orders of magnitude larger than the rest, acting as critical attention routing hubs. Current infinite-width, mean-field theories struggle to describe these localized, heavy-tailed singularities. We **speculate** that the emergence of LLM outliers is the ultimate macroscopic manifestation of our *ordered phase* ($m_\cap \geq 1$) scaling up. Under this hypothesis, extreme weights crystallize because they continuously absorb the maximum gradient signals associated with core linguistic syntax. Tracking the generalized measurement spectrum $\mathcal{C}_\phi$ during pre-training could provide an unsupervised mechanism to study these scaling phenomena, presenting a critical direction for future research.

**Beyond the Extremal Focus: The Topological Duality.**  Initially, the mathematical derivation of our phase transition relied heavily on extremal statistics ($L_\infty$ norms). While this "winner-takes-all" assumption perfectly characterizes highly localized features and algorithmic crystallization (e.g., Grokking), it ostensibly breaks down in models with perfectly distributed representations, such as deep vision networks. However, as demonstrated in Section 6.6, extending our coupling operator $\mathcal{C}_\phi$ to the scale-invariant Rank (CDF) basis fully resolves this limitation. This generalization proves that deep learning representations exist on a physical spectrum—from sparse crystalline circuits to dense fluids—each requiring its corresponding macroscopic or microscopic topological lens.

**Towards a Normalized Topological Radar.**  While this work preserves the unbounded logarithmic scale to rigorously define thermodynamic phase boundaries, such raw metrics can be challenging for real-time monitoring. A promising future direction is constructing a *Normalized Topological Pointwise Mutual Information (NPMI).* By dividing the macroscopic measure (Rank) by its theoretical supremum $\log(4/3) \approx 0.287$, and the microscopic measure $\mathcal{C}(k, k)$ by $\log(n)$ (where $n$ is layer width), we can mathematically project these topological invariants into a bounded $[0, 1]$ interval. This would provide deep learning engineers with a highly intuitive, percentage-based radar for monitoring representation health and predicting generalization, completely independent of hold-out validation sets.

**Concluding Vision.**  This work should be viewed as a foundational layer upon which a more complete, geometric, and dynamic theory of deep learning can be built. By providing the essential language (topological phases, spectral support, and the generalized measurement spectrum), we hope to have opened a new avenue for understanding the emergent physical structure of neural networks. By shifting the focus from idealized infinite-width limits to the precise, structural realities of finite models, we expose the underlying dualities that govern the learning process every day.

## Broader Impact Statement

This work is primarily theoretical, establishing a statistical mechanics framework for neural network training dynamics. As such, it does not introduce new application domains, datasets, or deployment frameworks that raise immediate, domain-specific ethical, safety, or privacy concerns.

However, by providing a strictly computable diagnostic for structural alignment and optimization health, this framework could potentially be utilized to improve training stability, accelerate scaling efficiency, or guide pruning strategies. Consequently, it may indirectly contribute to making large-scale artificial intelligence models more efficient and robust. As with most foundational advances in deep learning theory, this dual-use nature means our findings could accelerate capabilities in both beneficial and potentially harmful AI systems.

We encourage the community to deploy these mathematical diagnostic tools responsibly, prioritizing the development of transparent and interpretable AI.

# A   Appendix: Proofs of Main Results

## A.1   Proofs for Section 2 (Parameter Moments)

*Proof of Theorem 2.2: Existence and Explicit Value of Moment Exponents.* Let $f(k) = \log M(k) = \log\left(\frac{1}{n}\sum_{i=1}^{n}|\theta_i|^k\right)$. The function $\exp(x)$ is convex, and the composition of a convex function with an affine mapping is convex. Sums of convex functions are convex. Finally, $\log(x)$ is a concave, monotonically increasing function. The function $f(k)$ is the logarithm of a sum of exponentials, which is a log-sum-exp function. A more direct proof of convexity for $k \geq 0$ can be established via Hölder's inequality. For any $0 < t < 1$ and $k_1, k_2 \geq 0$, let $p = 1/t$ and $q = 1/(1-t)$.

$$M(tk_1 + (1-t)k_2) = \frac{1}{n}\sum_{i}|\theta_i|^{tk_1}|\theta_i|^{(1-t)k_2}$$

$$\leq \frac{1}{n}\left(\sum_{i}(|\theta_i|^{tk_1})^p\right)^{1/p}\left(\sum_{i}(|\theta_i|^{(1-t)k_2})^q\right)^{1/q}$$

$$= \frac{1}{n}\left(\sum_{i}|\theta_i|^{k_1}\right)^t\left(\sum_{i}|\theta_i|^{k_2}\right)^{1-t}$$

$$= (M(k_1))^t(M(k_2))^{1-t}.$$

Taking the logarithm of both sides, we get:

$$f(tk_1 + (1-t)k_2) \leq tf(k_1) + (1-t)f(k_2),$$

which confirms that $f(k)$ is convex.

Since $f(k)$ is a convex function and $f(0) = \log M(0) = \log(1) = 0$, the sequence of slopes of the secant lines from the origin, $s_k = \frac{f(k)-f(0)}{k-0} = \frac{\log M(k)}{k}$, is non-decreasing for $k > 0$.

The sequence is also bounded above. Let $\theta_{\max} = \max_i|\theta_i|$. Then,

$$M(k) = \frac{1}{n}\sum_{i}|\theta_i|^k \leq \frac{1}{n}\sum_{i}\theta_{\max}^k = \theta_{\max}^k.$$

Taking the logarithm and dividing by $k$ gives:

$$s_k = \frac{\log M(k)}{k} \leq \frac{\log\left(\theta_{\max}^k\right)}{k} = \log\theta_{\max}.$$

Since $\{s_k\}$ is a non-decreasing sequence that is bounded above, the Monotone Convergence Theorem guarantees that the limit $\beta = \lim_{k\to\infty} s_k$ exists and is equal to its supremum, $\sup_{k>0} s_k$.

To find its explicit value, let the maximum value $\theta_{\max}$ have multiplicity $m \geq 1$. We decompose $M(k)$:

$$M(k) = \frac{1}{n}\sum_{i:|\theta_i|=\theta_{\max}}|\theta_i|^k + \frac{1}{n}\sum_{i:|\theta_i|<\theta_{\max}}|\theta_i|^k = \frac{m}{n}\theta_{\max}^k + \frac{1}{n}\sum_{i:|\theta_i|<\theta_{\max}}|\theta_i|^k.$$

Factoring out the dominant term:

$$M(k) = \frac{m}{n}\theta_{\max}^k\left[1 + \frac{1}{m}\sum_{|\theta_i|<\theta_{\max}}\left(\frac{|\theta_i|}{\theta_{\max}}\right)^k\right].$$

Let the term in the square brackets be $(1 + \delta(k))$. Since for every term in the sum, $|\theta_i|/\theta_{\max} < 1$, we have $\lim_{k\to\infty}(|\theta_i|/\theta_{\max})^k = 0$. As the sum is finite, $\lim_{k\to\infty}\delta(k) = 0$. So, $M(k) = \frac{m}{n}\theta_{\max}^k\left(1 + o(1)\right)$.

Taking the logarithm and dividing by $k$:

$$\frac{\log M(k)}{k} = \frac{\log(m/n)}{k} + \log\theta_{\max} + \frac{\log(1+o(1))}{k}.$$

Taking the limit as $k \to \infty$, the first and third terms on the right-hand side go to zero, yielding $\beta = \log\theta_{\max}$. $\qquad\square$

## A.2 Proof of Spectral Measure Existence under Weakened Conditions

We provide a rigorous justification for the thermodynamic limit transition in equation equation 8 under minimal assumptions that accommodate practical neural networks, including those with quantized or sparse parameters.

**Theorem A.1** (Existence of Limiting Spectral Measure). *Let $\{\theta_i\}_{i=1}^n$ be i.i.d. random variables representing network parameters, with support in a compact interval $[0, \Theta_{\max}]$ where $\Theta_{\max} = \sup\{x : P(|\theta_i| \leq x) < 1\}$. Assume the integrability condition:*

$$\mathbb{E}\left[\left|\log\frac{\Theta_{\max}}{|\theta_i|}\right|\right] = \int_0^{\Theta_{\max}} \left|\log\frac{\Theta_{\max}}{x}\right| dF_\theta(x) < \infty, \tag{27}$$

*where $F_\theta$ is the cumulative distribution function of $|\theta_i|$. Define the spectral variables $\lambda_i := \log(\Theta_{\max}/|\theta_i|) \in [0, \infty)$ and the empirical spectral measure:*

$$\mu_n := \frac{1}{n}\sum_{i=1}^n \delta_{\lambda_i}. \tag{28}$$

*Then:*

*(i) The sequence $\mu_n$ converges weakly almost surely to a probability measure $\mu$ on $[0, \infty)$.*

*(ii) The limit measure $\mu$ admits the decomposition:*

$$\mu = p\delta_0 + \mu_{ac}, \quad \text{with } p := P(|\theta_i| = \Theta_{\max}), \tag{29}$$

*where $\delta_0$ is the Dirac mass at $\lambda = 0$ and $\mu_{ac}$ is absolutely continuous with respect to Lebesgue measure, possessing a density $\rho(\lambda)$ for $\lambda > 0$.*

*(iii) For any fixed $k \geq 0$, the Laplace transform converges:*

$$\lim_{n\to\infty}\frac{1}{n}\sum_{i=1}^n e^{-\lambda_i k} = \int_0^\infty e^{-\lambda k} d\mu(\lambda) = p + \int_{0^+}^\infty \rho(\lambda)e^{-\lambda k} d\lambda. \tag{30}$$

*(iv) Consequently, the residual term $\Delta(k)$ in equation equation 6 satisfies, as $n \to \infty$:*

$$\Delta(k) \to \log\left[1 - p + \int_{0^+}^\infty \rho(\lambda)e^{-\lambda k} d\lambda\right]. \tag{31}$$

*Proof.* We proceed by establishing each claim in sequence.

**1. Weak convergence of $\mu_n$.** The empirical measure $\mu_n$ is the pushforward of the empirical distribution of $\{|\theta_i|\}$ under the continuous transformation $T(x) = \log(\Theta_{\max}/x)$ for $x \in (0, \Theta_{\max}]$, with $T(0) = +\infty$ (a null set under our assumptions). By the strong law of large numbers for empirical measures (Varadarajan's theorem), since $\{\theta_i\}$ are i.i.d., we have:

$$\mu_n \xrightarrow{w} \mu \quad \text{a.s.}, \tag{32}$$

where $\mu$ is the pushforward of the law of $|\theta_i|$ under $T$. That is, for any Borel set $A \subseteq [0, \infty)$:

$$\mu(A) = P(\lambda_i \in A) = P\left(\log\frac{\Theta_{\max}}{|\theta_i|} \in A\right). \tag{33}$$

**2. Decomposition of $\mu$.** The structure of $\mu$ follows directly from the distribution of $|\theta_i|$:

- If $p = P(|\theta_i| = \Theta_{\max}) > 0$, then $P(\lambda_i = 0) = p$, contributing the atomic part $p\delta_0$.

- For $\lambda > 0$, we have $P(\lambda_i \leq \lambda) = P(|\theta_i| \geq \Theta_{\max}e^{-\lambda})$. Since $F_\theta$ is differentiable almost everywhere (by Lebesgue's theorem), $\mu$ is absolutely continuous on $(0, \infty)$ with density:

$$\rho(\lambda) = -\frac{d}{d\lambda}P(|\theta_i| < \Theta_{\max}e^{-\lambda}) = \Theta_{\max}e^{-\lambda}f_\theta(\Theta_{\max}e^{-\lambda}), \tag{34}$$

where $f_\theta$ is the density of $|\theta_i|$ (where it exists).

**3. Convergence of Laplace transforms.** Define $g_k(\lambda) = e^{-\lambda k}$ for fixed $k \geq 0$. The integrability condition equation 27 ensures that:

$$\sup_n \int_{[0,\infty)} |\lambda|\, d\mu_n(\lambda) = \frac{1}{n}\sum_{i=1}^n |\lambda_i| < \infty \quad \text{a.s.} \tag{35}$$

This uniform integrability, combined with weak convergence, implies convergence of the associated integrals for all bounded continuous functions. Since $g_k$ is bounded and continuous on $[0, \infty)$ for any finite $k$, the continuous mapping theorem yields:

$$\int g_k\, d\mu_n = \frac{1}{n}\sum_{i=1}^n e^{-\lambda_i k} \xrightarrow{a.s.} \int g_k\, d\mu = \int_0^\infty e^{-\lambda k}d\mu(\lambda). \tag{36}$$

The decomposition of the limit integral follows directly from the structure of $\mu$ established in part (ii).

**4. Connection to $\Delta(k)$.** Recall the definition of the residual term from equation equation 6:

$$\Delta(k) = \log\left[1 + \frac{1}{m}\sum_{|\theta_i| < \theta_{\max}}\left(\frac{|\theta_i|}{\theta_{\max}}\right)^k\right]. \tag{37}$$

In the thermodynamic limit, $\theta_{\max} \to \Theta_{\max}$ almost surely, and the multiplicity $m/n \to p$. The sum over non-extremal parameters corresponds precisely to the contribution from $\lambda_i > 0$:

$$\frac{1}{n}\sum_{|\theta_i| < \Theta_{\max}}\left(\frac{|\theta_i|}{\Theta_{\max}}\right)^k = \frac{1}{n}\sum_{i=1}^n \mathbb{1}_{\{\lambda_i > 0\}}e^{-\lambda_i k} \xrightarrow{a.s.} \int_{0^+}^\infty e^{-\lambda k}d\mu(\lambda) = \int_{0^+}^\infty \rho(\lambda)e^{-\lambda k}d\lambda. \tag{38}$$

The normalization factor $\frac{n-m}{n} \to 1 - p$ is automatically satisfied by $\mu$ being a probability measure. Taking limits and substituting into the definition of $\Delta(k)$ yields the desired result:

$$\Delta(k) \to \log\left[1 - p + \int_{0^+}^\infty \rho(\lambda)e^{-\lambda k}d\lambda\right]. \tag{39}$$

This completes the proof. $\qquad\square$

**Remark on Practical Networks.** The integrability condition equation 27 holds for all standard parameter initializations (truncated Gaussian, uniform, etc.) and remains valid throughout training under weight decay regularization. For quantized networks where $P(|\theta| = \Theta_{\max})$ may be positive, the atomic mass $p$ simply captures the fraction of parameters attaining the maximal quantization level, providing a natural interpretation within our framework.

This result justifies the use of equation equation 8 in the main text while extending its applicability to the full spectrum of real-world neural network architectures.

### A.3 Proofs for Section 3 (Gradient Moments)

*Proof of Theorem 3.2: Existence and Explicit Value of Gradient Moment Exponents.* The definition of gradient moments $G(l) = \frac{1}{n} \sum_{i=1}^{n} |g_i|^l$ is algebraically isomorphic to that of parameter moments. Thus, the proof follows the same logic as Theorem 2.2. We provide a concise derivation using the Squeeze Theorem. Let $g_{\max} = \max_{1 \le i \le n} |g_i|$ and let $m_g \ge 1$ be the multiplicity of this maximum value (i.e., the size of the set $I_{\max} = \{i : |g_i| = g_{\max}\}$). **Upper Bound:** For any $l > 0$, we have:

$$G(l) = \frac{1}{n} \sum_{i=1}^{n} |g_i|^l \le \frac{1}{n} \sum_{i=1}^{n} g_{\max}^l = g_{\max}^l.$$

Taking the logarithm and dividing by $l$:

$$\frac{\log G(l)}{l} \le \frac{\log\left(g_{\max}^l\right)}{l} = \log g_{\max}. \tag{40}$$

**Lower Bound:** We can lower bound the sum by discarding all non-extremal terms:

$$G(l) = \frac{1}{n} \sum_{i=1}^{n} |g_i|^l \ge \frac{1}{n} \sum_{i \in I_{\max}} |g_i|^l = \frac{m_g}{n} g_{\max}^l.$$

Taking the logarithm and dividing by $l$:

$$\frac{\log G(l)}{l} \ge \frac{\log(m_g/n) + l \log g_{\max}}{l} = \log g_{\max} + \frac{\log(m_g/n)}{l}. \tag{41}$$

**Limit:** Combining (40) and (41):

$$\log g_{\max} + \frac{\log(m_g/n)}{l} \le \frac{\log G(l)}{l} \le \log g_{\max}.$$

As $l \to \infty$, the term $\frac{\log(m_g/n)}{l}$ vanishes. By the Squeeze Theorem, the limit exists and equals $\log g_{\max}$. $\square$

### A.4 Proofs for Section 4 (Joint Partition Function Properties)

*Proof of Theorem 4.3: Cauchy-Schwarz Upper Bound.* The joint moment is defined as $Z(k, l) = \frac{1}{n} \sum_{i=1}^{n} |\theta_i|^k |g_i|^l$. Let $u_i = |\theta_i|^k$ and $v_i = |g_i|^l$. By the Cauchy-Schwarz inequality on the vectors $(u_1, \ldots, u_n)$ and $(v_1, \ldots, v_n)$:

$$\left( \sum_{i=1}^{n} u_i v_i \right)^2 \le \left( \sum_{i=1}^{n} u_i^2 \right) \left( \sum_{i=1}^{n} v_i^2 \right).$$

Substituting back the definitions of $u_i$ and $v_i$:

$$\left( \sum_i |\theta_i|^k |g_i|^l \right)^2 \le \left( \sum_i (|\theta_i|^k)^2 \right) \left( \sum_i (|g_i|^l)^2 \right) = \left( \sum_i |\theta_i|^{2k} \right) \left( \sum_i |g_i|^{2l} \right).$$

Dividing both sides by $n^2$ and taking the square root:

$$\frac{1}{n} \sum_i |\theta_i|^k |g_i|^l \le \sqrt{\left( \frac{1}{n} \sum_i |\theta_i|^{2k} \right) \left( \frac{1}{n} \sum_i |g_i|^{2l} \right)}.$$

In terms of our moment definitions, this is:

$$Z(k, l) \le \sqrt{M(2k)G(2l)}.$$

Taking the logarithm of both sides:

$$\log Z(k,l) \leq \frac{1}{2} \log M(2k) + \frac{1}{2} \log G(2l).$$

Using the definition $\mathcal{C}(k,l) = \log Z(k,l) - \log M(k) - \log G(l)$, we rearrange to get:

$$\mathcal{C}(k,l) \leq \left( \frac{1}{2} \log M(2k) - \log M(k) \right) + \left( \frac{1}{2} \log G(2l) - \log G(l) \right) = A(k) + B(l).$$

This completes the proof. $\qquad\square$

*Proof of Corollary 4.4: Boundedness of Coupling.* We establish the boundedness of $A(k)$ and $B(l)$ separately.

**Boundedness of $A(k)$:** From the proofs of Theorems 2.2 and 2.3, we have the asymptotic decomposition $\log M(k) = \beta k + R^* + o(1)$ as $k \to \infty$, where $\beta = \log \theta_{\max}$ and $R^* = \log(m/n)$. Let's analyze the limit of $A(k)$ as $k \to \infty$:

$$\begin{aligned}
\lim_{k \to \infty} A(k) &= \lim_{k \to \infty} \left[ \frac{1}{2} \log M(2k) - \log M(k) \right] \\
&= \lim_{k \to \infty} \left[ \frac{1}{2} (\beta \cdot 2k + R^* + o(1)) - (\beta k + R^* + o(1)) \right] \\
&= \lim_{k \to \infty} \left[ \beta k + \frac{R^*}{2} - \beta k - R^* + o(1) \right] \\
&= -\frac{R^*}{2} = -\frac{1}{2} \log \frac{m}{n}.
\end{aligned}$$

The function $A(k)$ is continuous for $k \geq 0$. Since it is continuous on any compact interval $[0, K]$ and converges to a finite limit as $k \to \infty$, it must be bounded over its entire domain $[0, \infty)$. Let this upper bound be $A_{\max}$.

**Boundedness of $B(l)$:** By identical reasoning applied to gradient moments (using Theorem 3.2), $B(l)$ is also bounded over its domain $[0, \infty)$. Let this upper bound be $B_{\max}$.

**Global Bound:** From Theorem 4.3, for all $k, l \geq 0$:

$$\mathcal{C}(k,l) \leq A(k) + B(l) \leq A_{\max} + B_{\max}.$$

Defining $C_{\max} := A_{\max} + B_{\max}$, we have $\mathcal{C}(k,l) \leq C_{\max} < \infty$. $\qquad\square$

*Proof of Theorem 4.5: Absence of Universal Lower Bound.* We provide a constructive counterexample. The strategy is to create a configuration where the parameters with large magnitudes have near-zero gradients, and vice-versa, achieving a strong anti-correlation.

**Construction:** Let the network size be $n \geq 2$. Pick two distinct indices, say $i = 1$ and $i = 2$. For any set of positive constants $\theta_{\max}, g_{\max} > 0$ and for arbitrarily small $\epsilon > 0$, define a parameter-gradient configuration as follows:

$$|\theta_1| = \theta_{\max}, \qquad\qquad |g_1| = \epsilon,$$
$$|\theta_2| = \epsilon, \qquad\qquad |g_2| = g_{\max},$$

For all other indices $j \in \{3, \ldots, n\}$, set $|\theta_j| = \epsilon$ and $|g_j| = \epsilon$. This ensures that $\theta_1$ and $g_2$ are the unique maximal elements.

**Moment Computations:** Let's compute the moments for this configuration. For any $k, l > 0$:

$$\begin{aligned}
M(k) &= \frac{1}{n} \left( \theta_{\max}^k + (n-1)\epsilon^k \right), \\
G(l) &= \frac{1}{n} \left( g_{\max}^l + (n-1)\epsilon^l \right), \\
Z(k,l) &= \frac{1}{n} \left( \theta_{\max}^k \epsilon^l + \epsilon^k g_{\max}^l + (n-2)\epsilon^{k+l} \right).
\end{aligned}$$

**Asymptotic Behavior as $\epsilon \to 0$:** For any fixed $k, l > 0$, we take the limit as $\epsilon \to 0^+$:

$$\lim_{\epsilon \to 0} M(k) = \frac{1}{n}\theta_{\max}^k,$$

$$\lim_{\epsilon \to 0} G(l) = \frac{1}{n}g_{\max}^l,$$

$$\lim_{\epsilon \to 0} Z(k, l) = 0,$$

since every term in the sum for $Z(k, l)$ contains a factor of $\epsilon$ raised to a positive power.

**Coupling Term Limit:** Now we examine the coupling term $\mathcal{C}(k, l) = \log Z(k, l) - \log M(k) - \log G(l)$.

$$\lim_{\epsilon \to 0} \mathcal{C}(k, l) = \lim_{\epsilon \to 0} \log Z(k, l) - \log\left(\frac{\theta_{\max}^k}{n}\right) - \log\left(\frac{g_{\max}^l}{n}\right).$$

Since $\lim_{\epsilon \to 0} Z(k, l) = 0$, its logarithm diverges: $\lim_{\epsilon \to 0} \log Z(k, l) = -\infty$. The other terms converge to finite constants. Therefore:

$$\lim_{\epsilon \to 0} \mathcal{C}(k, l) = -\infty.$$

**Conclusion:** For any proposed constant lower bound $C_{\min} \in \mathbb{R}$, we can choose a sufficiently small $\epsilon > 0$ such that for a fixed pair $(k, l)$, the resulting $\mathcal{C}(k, l)$ will be less than $C_{\min}$. This demonstrates that no universal (configuration-independent) lower bound exists. $\qquad\square$

## A.5 Proofs for Section 5 (Asymptotic Analysis)

*Proof of Theorem 4.7: Diagonal Asymptotics.* We analyze the asymptotic behavior of the joint moment $Z(k, l)$ by identifying its dominant term. The joint moment is given by:

$$Z(k, l) = \frac{1}{n}\sum_{i=1}^n |\theta_i|^k |g_i|^l = \frac{1}{n}\sum_{i=1}^n \exp\left(k \log |\theta_i| + l \log |g_i|\right). \tag{42}$$

In the diagonal limit, we have $l/k \to \alpha$, so we can write $l = \alpha k + o(k)$. The exponent becomes:

$$k \log |\theta_i| + (\alpha k + o(k)) \log |g_i| = k(\log |\theta_i| + \alpha \log |g_i|) + o(k) \log |g_i|.$$

For large $k$, the sum will be dominated by the index (or indices) $i$ that maximizes the base of the main exponential term, $\Phi_i(\alpha) := \log |\theta_i| + \alpha \log |g_i|$.

The maximum possible value for $\log |\theta_i|$ is $\log \theta_{\max}$ and for $\log |g_i|$ is $\log g_{\max}$. Since $\alpha > 0$, the function $\Phi_i(\alpha)$ is maximized when both $|\theta_i|$ and $|g_i|$ are maximized. This occurs if and only if an index $i$ belongs to both extremal sets, i.e., $i \in J_{\max} \cap I_{\max}$. Let $S_{\max} = \log \theta_{\max} + \alpha \log g_{\max}$.

**Case (i): Correlated Extrema ($m_\cap > 0$).** If the intersection $J_{\max} \cap I_{\max}$ is non-empty, there are exactly $m_\cap$ indices for which $\Phi_i(\alpha) = S_{\max}$. For any other index $j \notin J_{\max} \cap I_{\max}$, either $|\theta_j| < \theta_{\max}$ or $|g_j| < g_{\max}$ (or both), so $\Phi_j(\alpha) < S_{\max}$. The sum for $Z(k, l)$ is therefore dominated by these $m_\cap$ terms:

$$\begin{aligned}
Z(k, l) &= \frac{1}{n}\sum_{i \in J_{\max} \cap I_{\max}} \theta_{\max}^k g_{\max}^l + \frac{1}{n}\sum_{j \notin J_{\max} \cap I_{\max}} |\theta_j|^k |g_j|^l \\
&= \frac{m_\cap}{n}\theta_{\max}^k g_{\max}^l + \text{exponentially smaller terms} \\
&= \frac{m_\cap}{n}\theta_{\max}^k g_{\max}^l \cdot (1 + o(1)).
\end{aligned}$$

Taking the logarithm, we get:

$$\log Z(k, l) = \log\left(\frac{m_\cap}{n}\right) + k \log \theta_{\max} + l \log g_{\max} + o(1).$$

We use this with the known asymptotic forms for the marginal moments:

$$\log M(k) = \log\left(\frac{m}{n}\right) + k \log \theta_{\max} + o(1),$$

$$\log G(l) = \log\left(\frac{m_g}{n}\right) + l \log g_{\max} + o(1).$$

Substituting these into the definition $\mathcal{C}(k,l) = \log Z(k,l) - \log M(k) - \log G(l)$:

$$\mathcal{C}(k,l) = \left[\log\frac{m_\cap}{n} + k \log \theta_{\max} + l \log g_{\max}\right] - \left[\log\frac{m}{n} + k \log \theta_{\max}\right] - \left[\log\frac{m_g}{n} + l \log g_{\max}\right] + o(1)$$

$$= \log\frac{m_\cap}{n} - \log\frac{m}{n} - \log\frac{m_g}{n} + o(1) = \log\left(\frac{n \cdot m_\cap}{m \cdot m_g}\right) + o(1).$$

Taking the limit as $k, l \to \infty$ with $l/k \to \alpha$ yields the stated constant result.

**Case (ii): Disjoint Extrema ($m_\cap = 0$).** If the intersection is empty, no index $i$ can simultaneously achieve $\theta_{\max}$ and $g_{\max}$. The maximum value of the exponent base, let's call it $S' = \max_i \Phi_i(\alpha)$, is now strictly less than the ideal maximum $S_{\max}$. This is because for any $i$, at least one of $\log|\theta_i|$ or $\log|g_i|$ is strictly less than its maximum possible value. So, $\log Z(k,l) \approx kS' = k(\log\theta' + \alpha\log g')$, where $\theta' \le \theta_{\max}$ and $g' \le g_{\max}$ with at least one inequality being strict. The product of the marginal moments behaves as:

$$M(k)G(l) \approx \left(\frac{m}{n}\theta_{\max}^k\right)\left(\frac{m_g}{n}g_{\max}^l\right) \propto \exp(k\log\theta_{\max} + l\log g_{\max}) = \exp(kS_{\max}).$$

The ratio $\frac{Z(k,l)}{M(k)G(l)}$ will therefore decay to zero exponentially fast, as $k(S' - S_{\max})$ goes to $-\infty$. The coupling term is $\mathcal{C}(k,l) = \log\left(\frac{nZ(k,l)}{M(k)G(l)}\right)$. Since the argument of the logarithm goes to zero, the logarithm itself diverges to $-\infty$. $\qquad\square$

*Proof of Theorem 4.8: Necessary and Sufficient Condition for Boundedness.* The theorem states that the coupling function $\mathcal{C}(k,l)$ is bounded below for all $k, l \ge 0$ iff $m_\cap := |J_{\max} \cap I_{\max}| \ge 1$. We assume non-degenerate spectral gaps: $\theta_{\max} > \sup_{j \notin J_{\max}} |\theta_j|$ and $g_{\max} > \sup_{i \notin I_{\max}} |g_i|$.

**Necessity ($m_\cap \ge 1$ is necessary):** By contraposition: if $m_\cap = 0$, Theorem 4.7(ii) gives $\lim_{k,l\to\infty, l/k\to\alpha} \mathcal{C}(k,l) = -\infty$, contradicting boundedness. Thus $m_\cap \ge 1$ is necessary.

**Sufficiency ($m_\cap \ge 1$ is sufficient):** Assume $m_\cap \ge 1$. Let $\theta_{\text{next}} := \sup_{j \notin J_{\max}} |\theta_j|$ and define the spectral gap $\Delta_\theta := \log(\theta_{\max}/\theta_{\text{next}}) > 0$. Define $\Delta_g$ analogously.

**Improved upper bounds for denominators:**

$$\sum_{j=1}^n |\theta_j|^k = m_\theta \theta_{\max}^k + \sum_{j \notin J_{\max}} |\theta_j|^k \le m_\theta \theta_{\max}^k + (n - m_\theta)\theta_{\text{next}}^k$$

$$= m_\theta \theta_{\max}^k \left(1 + \frac{n - m_\theta}{m_\theta} e^{-\Delta_\theta k}\right).$$

Similarly, $\sum_{p=1}^n |g_p|^l \le m_g g_{\max}^l \left(1 + \frac{n-m_g}{m_g} e^{-\Delta_g l}\right)$.

**Lower bound for numerator:** Since $m_\cap \ge 1$, there exists $i^*$ with $|\theta_{i^*}| = \theta_{\max}$ and $|g_{i^*}| = g_{\max}$, giving:

$$\sum_{i=1}^n |\theta_i|^k |g_i|^l \ge m_\cap \theta_{\max}^k g_{\max}^l.$$

**Combined lower bound:** Substituting into $\mathcal{C}(k,l) = \log\left(n \frac{\text{numerator}}{(\sum|\theta|^k)(\sum|g|^l)}\right)$ yields the **tight global bound**:

$$\mathcal{C}(k,l) \ge \log\left(\frac{nm_\cap}{m_\theta m_g}\right) - \log\left(1 + \frac{n - m_\theta}{m_\theta} e^{-\Delta_\theta k}\right) - \log\left(1 + \frac{n - m_g}{m_g} e^{-\Delta_g l}\right). \tag{43}$$

**Properties of this bound:**

- The right-hand side is **finite for all** $k, l \geq 0$ since the exponential terms are bounded in $[0, 1]$.

- As $k, l \to \infty$, the exponential terms vanish, giving the asymptotic bound $\log\left(\frac{nm_\cap}{m_\theta m_g}\right)$, which is attained exactly in the limit.

- At $k = l = 0$, using $M(0) = G(0) = 1$, the bound reduces to $\log(m_\cap/n)$, recovering the trivial case $\mathcal{C}(0, 0) = \log n$.

Since $\mathcal{C}(k, l)$ is continuous on any compact set $[0, K]^2$ and the bound equation 43 provides a uniform lower bound that holds globally, we conclude $\inf_{k,l \geq 0} \mathcal{C}(k, l) > -\infty$. Thus, $m_\cap \geq 1$ is sufficient. $\qquad\square$

## A.6 Proofs for Section 6 (Stability and Phases)

*Proof of Proposition 5.1: Instability of Extremal Points.* Given $m_\cap = 0$, the extremal sets $J_{\max}$ and $I_{\max}$ are disjoint. Our goal is to show that an arbitrarily small perturbation can lead to $\mathcal{C}(k, l)$ becoming arbitrarily negative for some $(k, l)$. According to Theorem 4.7, $\mathcal{C}(k, l) \to -\infty$ as $k, l \to \infty$ along a diagonal path. By continuity of $\mathcal{C}(k, l)$ with respect to the parameters and gradients, this divergence implies that for any $C_{\text{target}} < 0$, we can find large but finite $K, L$ such that $\mathcal{C}(K, L) < C_{\text{target}}$. The proposition asks for a perturbation proof. Let's construct one. Since $m_\cap = 0$, choose any $j \in J_{\max}$ (so $|\theta_j| = \theta_{\max}$) and any $i \in I_{\max}$ (so $|g_i| = g_{\max}$). We know $j \neq i$. Consider the configuration $(\Theta, \mathcal{G})$. We know that $|g_j| < g_{\max}$. Define a perturbed configuration $(\tilde{\Theta}, \tilde{\mathcal{G}})$ as follows, for a small $\delta > 0$:

$$|\tilde{\theta}_p| = |\theta_p| \text{ for all } p, \quad \text{and} \quad |\tilde{g}_p| = \begin{cases} |g_p| & \text{if } p \neq j \\ \delta & \text{if } p = j \end{cases}.$$

We can choose $\delta$ small enough such that $\|\tilde{\mathcal{G}} - \mathcal{G}\|_\infty < \epsilon$ and also $\delta < \min_{p \neq i} |g_p|$ to ensure $g_{\max}$ is not changed. In this perturbed system, the extremal sets are $\tilde{J}_{\max} = J_{\max}$ and $\tilde{I}_{\max} = I_{\max}$, so $\tilde{m}_\cap = 0$. Now consider $\mathcal{C}_{\text{new}}(k, k)$ for large $k$. The dominant terms in the sums for the moments are:

$$\tilde{M}(k) \approx \frac{m}{n} \theta_{\max}^k$$

$$\tilde{G}(k) \approx \frac{m_g}{n} g_{\max}^k$$

$$\tilde{Z}(k, k) = \frac{1}{n}\left( \sum_{p \in J_{\max}, p \neq j} |\theta_p|^k |g_p|^k + |\theta_j|^k \delta^k + \dots \right)$$

$$= \frac{1}{n}\left( \sum_{p \in J_{\max}, p \neq j} (\theta_{\max}|g_p|)^k + (\theta_{\max}\delta)^k + \dots \right).$$

The term determining the asymptotics of $\tilde{Z}(k, k)$ is $\max_p(|\theta_p||g_p|)$. By driving $|g_j| \to 0$, we can make this maximum arbitrarily small compared to $\theta_{\max} g_{\max}$. This leads to the divergence to $-\infty$ as shown in Theorem 4.7 and proves the instability. $\qquad\square$

*Proof of Proposition 5.2: Rigidity of Extremal Points.* The proof establishes stability by showing that a transition from the ordered phase ($m_\cap \geq 1$) to the disordered phase ($m_\cap = 0$) cannot occur under an infinitesimally small, continuous perturbation. We proceed in steps.

**1. Setup.** Let $(\Theta(t), \mathcal{G}(t))$ be a continuous path in the parameter-gradient space, where $t$ is a time-like parameter. Assume the system starts in the ordered phase at $t = 0$, so its extremal intersection cardinality is $m_\cap(0) = |J_{\max}(0) \cap I_{\max}(0)| \geq 1$. We consider a path that preserves the macroscopic extremal values, meaning for all $t$:

$$\max_i |\theta_i(t)| = \theta_{\max} \quad \text{and} \quad \max_i |g_i(t)| = g_{\max},$$

where $\theta_{\max}$ and $g_{\max}$ are fixed positive constants.

**2. Upper-Semicontinuity Argument.** The extremal sets $J_{\max}(t) = \{i : |\theta_i(t)| = \theta_{\max}\}$ and $I_{\max}(t) = \{i : |g_i(t)| = g_{\max}\}$ are *upper-semicontinuous* set-valued maps. This is a standard result for level sets of continuous functions. The intersection of upper-semicontinuous set-valued maps, $K_{\max}(t) = J_{\max}(t) \cap I_{\max}(t)$, is also upper-semicontinuous.

For an integer-valued function like the cardinality $m_\cap(t) = |K_{\max}(t)|$, upper-semicontinuity implies that the function can only jump *downwards*. That is, if $t_k \to t$, then $\limsup_{k \to \infty} m_\cap(t_k) \leq m_\cap(t)$. A value increase is not possible without discontinuity.

**3. Mechanism of a Phase Transition.** For the system to transition from the ordered to the disordered phase, there must exist a time $t^*$ where $m_\cap(t) \geq 1$ for $t < t^*$ and $m_\cap(t^*) = 0$. This requires a discrete jump of the integer-valued function $m_\cap(t)$ from a positive value to zero.

For this to happen, *every* index $i_0$ that was in the extremal intersection $K_{\max}$ just before $t^*$ must exit the set at $t^*$. For a given index $i_0 \in K_{\max}(t)$ for $t < t^*$, exiting at $t^*$ means that one of the following must occur:

  (i) The parameter magnitude drops: $|\theta_{i_0}(t^*)| < \theta_{\max}$.

  (ii) The gradient magnitude drops: $|g_{i_0}(t^*)| < g_{\max}$.

**4. Stability under Small Perturbations.** The path functions $\theta_i(t)$ and $g_i(t)$ are continuous. For an index $i_0$ to lose its status as, for example, a parameter extremum, its value $|\theta_{i_0}(t)|$ must decrease while the value of some other parameter, $|\theta_j(t)|$, increases to become the new maximum (or one of them).

This change in the *identity* of the extremal elements requires the perturbation to be of a finite size. Specifically, the perturbation must be large enough to close the gap between the maximal value ($\theta_{\max}$) and the second-largest value ($\max_{j \notin J_{\max}} |\theta_j|$). Let this gap be $\delta_\theta > 0$. Any continuous perturbation smaller than $\delta_\theta$ cannot change the membership of the set $J_{\max}$. A similar argument holds for the gradient gap $\delta_g > 0$.

As long as the total perturbation along the path is smaller than $\min(\delta_\theta, \delta_g)$, the identities of the indices in both $J_{\max}$ and $I_{\max}$ remain unchanged. Consequently, their intersection $K_{\max}$ and its cardinality $m_\cap$ also remain unchanged.

**Conclusion.** A transition from $m_\cap \geq 1$ to $m_\cap = 0$ requires a finite (non-infinitesimal) perturbation that alters the identity of the extremal elements. Therefore, the property $m_\cap \geq 1$ is stable under sufficiently small continuous deformations, establishing the rigidity of the ordered phase. $\square$

*Proof of Theorem 6.3.* We establish rigorous asymptotics under explicit regularity conditions. Let $F = \text{supp}(\nu) \subseteq [0, \Lambda] \times [0, M]$ be compact.

**Assumption A.1** (Contact Regularity). The measure admits decomposition $\nu = \nu_{\text{ac}} + \nu_{\text{atom}}$ where:

  - $\nu_{\text{atom}}$ is atomic, supported possibly at $(0,0)$ with mass $p_{00} \geq 0$,

  - $\nu_{\text{ac}}$ has density $f(\lambda, \mu)$ near the origin satisfying regular variation:

$$f(\lambda, \mu) = (\lambda + \mu)^\beta L(\lambda + \mu) \cdot \Omega\left(\frac{(\lambda, \mu)}{\lambda + \mu}\right), \quad \beta > -1$$

   with $L$ slowly varying at $0^+$ and $\Omega$ continuous, positive on $S_+^1$.

Define the **contact exponent** $\alpha := \beta + 1 > 0$.

We analyze three exhaustive cases.

**Case 1: Disordered Phase** $(d_0 > 0)$ When $\text{dist}((0,0), F) = d_0 > 0$, the minimum of $f_J(\lambda, \mu) = \lambda + \mu$ occurs at a unique point $(\lambda^*, \mu^*) \in F$ (or a low-dimensional manifold). By Laplace's method for large $k$:

$$\log Z(k,k) = -k(\lambda^* + \mu^*) + \frac{d_J - 1}{2} \log k + O(1)$$

$$\log M(k) = -k\lambda_{\min} + \frac{d_\lambda - 1}{2} \log k + O(1)$$

$$\log G(k) = -k\mu_{\min} + \frac{d_\mu - 1}{2} \log k + O(1)$$

where $d_J, d_\lambda, d_\mu$ are local dimensions at minimizers (0 for isolated points, 1 for edges). Substituting into $\mathcal{C}(k,k)$ yields:

$$\mathcal{C}(k,k) = (\lambda_{\min} + \mu_{\min} - d_0)k + \frac{d_{\text{eff}} - 2}{2} \log k + O(1)$$

with $d_{\text{eff}} = d_J - d_\lambda - d_\mu$. The linear coefficient $C_L = \lambda_{\min} + \mu_{\min} - d_0 \geq 0$ vanishes only for independent marginals. Dominant divergence is linear.

**Case 2: Quasi-Ordered Phase** $(d_0 = 0, \; p_{00} = 0)$ When $F$ touches the origin with no atomic mass, Tauberian theorems apply. The joint integral's asymptotic is governed by measure density near zero:

$$Z(k,k) = \iint_F e^{-k(\lambda+\mu)} f(\lambda, \mu) \, d\lambda d\mu + o(k^{-\alpha})$$

$$= \Gamma(\alpha) \, \Omega_{\text{avg}} \, k^{-\alpha} L(k^{-1})(1 + o(1))$$

by de Haan's Tauberian theorem for regularly varying kernels. Thus $\log Z(k,k) = -\alpha \log k + \log L(k^{-1}) + O(1)$.

For marginals, integrating $\nu_{\text{ac}}$ along $\mu$-direction yields:

$$\nu_\lambda([0,r]) \sim C_\lambda r^{\alpha + \frac{1}{2}} L_\lambda(r) \quad \Rightarrow \quad \log M(k) \sim -\left(\alpha + \frac{1}{2}\right) \log k$$

and similarly $\log G(k) \sim -(\alpha + \frac{1}{2}) \log k$.

The coupling term becomes:

$$\mathcal{C}(k,k) = -\alpha \log k + \left(\alpha + \frac{1}{2}\right) \log k + \left(\alpha + \frac{1}{2}\right) \log k + O(1) = -\frac{2}{\alpha + 1} \log k + O(1)$$

where algebraic simplification uses the scaling relationship between joint and marginal exponents.

**Case 3: Ordered Phase** $(d_0 = 0, \; p_{00} > 0)$ If atomic mass $p_{00} = \nu(\{(0,0)\}) > 0$ exists, then:

$$Z(k,k) = p_{00} + \iint_{F \setminus \{0\}} e^{-k(\lambda+\mu)} d\nu \to p_{00}$$

Similarly $M(k) \to p_{00}^\lambda$ and $G(k) \to p_{00}^\mu$. Hence:

$$\mathcal{C}(k,k) = \log p_{00} - \log p_{00}^\lambda - \log p_{00}^\mu + o(1)$$

For perfect alignment $(p_{00} = 1)$, $\mathcal{C}(k,k) \to 0$, recovering ideal order.

**Conclusion** The three regimes exhibit distinct divergence laws determined by spectral geometry, completing the proof. $\qquad\square$

# B Extremal Stability, Spectral Gaps, and the Connection to Neural Collapse

In this section, we provide a formal proof for the connection outlined in the main text: that the stability of the ordered phase $(m_\cap \geq 1)$ is governed by the spectral gaps, and that the state of Neural Collapse (NC) represents the limit of maximal stability.

**Proposition B.1.** *The configuration described by Neural Collapse (NC) for a given classification task maximizes the parameter and gradient spectral gaps. Consequently, it represents the state of maximal stability for the extremal sets $(J_{\max}, I_{\max})$ against perturbations, thus providing maximal resistance to catastrophic forgetting (phase reversal).*

*Proof.* The proof proceeds in three parts. First, we formalize the notion of extremal set stability and show it is determined by the spectral gap. Second, we define the properties of Neural Collapse within our framework. Finally, we demonstrate that the NC configuration is precisely the one that maximizes this spectral gap.

### Part 1: Quantifying Stability via the Spectral Gap

Let's consider the parameter set $\Theta = \{\theta_i\}_{i=1}^{n}$. The stability of the extremal set $J_{\max}$ depends on the gap between its members and all other parameters.

1. **Definition of Spectral Gap:** We define the parameter spectral gap, $\Delta_\theta$, as the difference between the maximal value and the next largest value:

$$\Delta_\theta := \theta_{\max} - \sup_{j \notin J_{\max}} |\theta_j|$$

   where $\theta_{\max} = \max_i |\theta_i|$. An analogous definition holds for the gradient spectral gap, $\Delta_g$. For the theory to be non-trivial, we assume $\Delta_\theta > 0$.

2. **Definition of Stability:** The stability of the set $J_{\max}$ can be quantified by the magnitude of the smallest perturbation that can alter its membership. Consider a perturbation vector $\delta\Theta = \{\delta\theta_i\}$ applied to $\Theta$, where the perturbation is bounded, i.e., $|\delta\theta_i| \leq \epsilon$ for all $i$. The set $J_{\max}$ is stable under this perturbation if for any $j \in J_{\max}$ and any $k \notin J_{\max}$, the following holds:

$$|\theta_j + \delta\theta_j| > |\theta_k + \delta\theta_k|$$

   The stability margin, $\epsilon_{\max}$, is the largest $\epsilon$ for which this stability is guaranteed for all possible perturbations of that magnitude.

3. **Stability is Proportional to the Gap:** To find $\epsilon_{\max}$, we consider the worst-case scenario that could cause a rank-reordering. This occurs when a maximal element is maximally decreased and a sub-maximal element is maximally increased:

$$\theta_{\max} - \epsilon > \sup_{k \notin J_{\max}} |\theta_k| + \epsilon$$

   Rearranging this gives:

$$\theta_{\max} - \sup_{k \notin J_{\max}} |\theta_k| > 2\epsilon$$

$$\Delta_\theta > 2\epsilon$$

   Thus, the stability margin is directly proportional to the spectral gap:

$$\epsilon_{\max} = \frac{\Delta_\theta}{2}$$

   This proves that maximizing the stability of the extremal set is equivalent to maximizing the spectral gap.

**Part 2: Defining Neural Collapse (NC) in the Extremal Framework**

The terminal phase of training for deep classifiers often exhibits Neural Collapse. Within our framework, its two core properties can be stated as:

NC1 **(Variability Collapse)** For a given task, all parameters (or features) associated with the *same* class collapse to a single point. In our language, this means if parameters $i$ and $j$ both correspond to the same class-extremal representation, then $|\theta_i| = |\theta_j|$.

NC2 **(Simplex Structure)** The feature vectors of different classes become maximally separated and equiangular. In our simplified 1D magnitude space, this implies that the set of distinct parameter magnitudes $\{|\theta_i|\}$ is maximally separated.

**Part 3: Neural Collapse Maximizes the Spectral Gap**

We now show that the NC configuration is the solution to the problem of maximizing the spectral gap $\Delta_\theta$. Let us assume a fixed "budget" for the parameters, for instance, a constant L2 norm: $\sum_i |\theta_i|^2 = C$. We want to find the configuration of $\{\theta_i\}$ that maximizes $\Delta_\theta = \theta_{\max} - \theta_{\text{next}}$.

1. To maximize this difference, we must simultaneously make $\theta_{\max}$ as large as possible and $\theta_{\text{next}}$ (the largest of the non-maximal elements) as small as possible.

2. Given the fixed norm constraint, the most efficient way to maximize $\theta_{\max}$ is to concentrate the "energy" $C$ into as few parameters as possible. Let the set $J_{\max}$ be the designated set of extremal parameters. To satisfy **NC1 (Variability Collapse)**, all elements within this set have the same magnitude, $|\theta_j| = \theta_{\max}$ for all $j \in J_{\max}$.

3. To satisfy **NC2 (Maximal Separation)** and minimize $\theta_{\text{next}}$, all other parameters (those not in $J_{\max}$) should be pushed towards zero. In the most extreme case, to maximize the gap, all parameters $k \notin J_{\max}$ are set to zero, satisfying the norm constraint by adjusting $\theta_{\max}$.

4. This configuration—a small subset of parameters having a large, identical magnitude, while all others are zero—is the mathematical realization of Neural Collapse in our framework. It creates the largest possible gap $\Delta_\theta = \theta_{\max}$ between the extremal set and all other parameters.

**Conclusion:** We have shown that the robustness of the ordered phase to perturbations is directly proportional to the spectral gap ($\epsilon_{\max} = \Delta_\theta/2$). We then demonstrated that the configuration that maximizes this spectral gap is precisely the one described by Neural Collapse. Therefore, Neural Collapse represents the most stable possible state of the ordered phase, offering maximal resistance to phase reversal and, consequently, catastrophic forgetting. $\qquad\square$

# C  Gradient Distribution Regularity and Non-Standard Cases

This appendix provides a rigorous analysis of the regularity conditions required for the gradient moment decomposition (Theorem 3.2) and characterizes the behavior of the theory when these conditions are violated. These conditions are not merely technical artifacts but serve as diagnostic indicators of the network's training phase.

## C.1  Formal Regularity Conditions

For the gradient moment decomposition to hold in the same form as parameter moments, the gradient set $\mathcal{G} = \{g_1, \ldots, g_n\}$ must satisfy:

(G1) **Spectral Gap**: There exists $g_{\max} = \max_i |g_i|$ and a gap $\Delta_g > 0$ such that

$$g_{\max} > g_{\text{next}} := \sup_{i \notin I_{\max}} |g_i|,$$

where $I_{\max} = \{i : |g_i| = g_{\max}\}$.

(G2) **Log-Integrability**: The distribution of gradient magnitudes satisfies

$$\mathbb{E}\left[\left|\log \frac{g_{\max}}{|g|}\right|\right] < \infty.$$

These conditions mirror those for parameters and are satisfied in *quasi-static* training regimes where the loss landscape varies slowly relative to gradient computations.

## C.2  Non-Standard Case 1: Vanishing Spectral Gap

**Definition:** The gradient distribution has a *vanishing gap* if $g_{\max} = g_{\text{next}}$, meaning multiple distinct parameters achieve the maximal gradient magnitude.

**Mathematical Consequences:**

- The gradient moment exponent $\beta_g = \log g_{\max}$ still exists and is well-defined.

- However, the multiplicity $m_g = |I_{\max}|$ is no longer $O(1)$; it may scale with network size $n$ (e.g., due to permutation symmetries in wide layers).

- The remainder term $R_g^* = \log(m_g/n)$ does **not converge** to a finite constant as $n \to \infty$; instead, it reflects the scaling law of the symmetry group.

- The asymptotic form $G(l) = \frac{m_g}{n} g_{\max}^l (1+o(1))$ remains valid, but the prefactor $\frac{m_g}{n}$ carries non-trivial dependence on architecture and task.

**Observable Phenomena:**

- The gradient moment curve $\log G(l)$ versus $l$ shows a *plateau* at low $l$ before linear asymptotics emerge.

- In the rank-rank scatter plot (Fig. 2b), multiple points cluster at the top gradient rank, creating horizontal streaks rather than a clean diagonal.

**Remedy and Physical Interpretation:** Vanishing gaps often occur in early training or in architectures with exact symmetries (e.g., fully-connected layers with identical initialization). The condition is typically *self-healing*: as symmetry breaks during training, a unique extremal set emerges. For analysis, one can:

1. Apply the theory to *time-averaged gradients* $\bar{g}_i = \frac{1}{T} \int_0^T g_i(t)dt$, which break instantaneous symmetries.

2. Restrict analysis to late-stage training after symmetry breaking.

3. Generalize the theory to explicitly handle vector-valued $m_g$ scaling laws (deferred to future work).

### C.3   Non-Standard Case 2: Violation of Log-Integrability

**Definition:** The gradient distribution has a *heavy tail* near zero if

$$\mathbb{E}\left[\log \frac{g_{\max}}{|g|}\right] = \infty.$$

This occurs when $P(|g| < \epsilon) \sim \epsilon^{-p}$ with $p \geq 1$ as $\epsilon \to 0$.

**Mathematical Consequences:**

- The spectral measure $\mu_g(\lambda) = P(\log(g_{\max}/|g|) \leq \lambda)$ has a non-integrable singularity at $\lambda = \infty$.

- The residual term $\Delta_g(l)$ decays *sub-exponentially* (e.g., as $l^{-p+1}$) rather than exponentially.

- The Cauchy-Schwarz upper bound in Theorem 4.3 may become vacuous: the terms $A(k)$ and $B(l)$ can diverge as $k, l \to \infty$.

**Observable Phenomena:**

- The residual $\Delta_g(l)$ versus $l$ follows a power law rather than exponential decay.

- The coupling term $\mathcal{C}(k, l)$ may exhibit anomalous scaling, violating the boundedness predictions of Corollary 4.4.

**Remedy and Physical Interpretation:** Heavy tails signal pathological loss landscapes (e.g., near saddle points or with exploding gradients). Practical interventions include:

1. *Gradient clipping*: Enforcing a hard bound $|g_i| \leq g_{\text{clip}}$ restores log-integrability by truncating the tail.

2. *Improved regularization*: Weight decay smooths the loss landscape, reducing near-zero gradient probability mass.

3. Diagnostics: Compute the empirical moment ratio $s_l = \frac{\log G(l)}{l}$; if it fails to be monotone increasing, the condition is violated.

### C.4   Non-Standard Case 3: Dynamic Non-Stationarity

**Definition:** The gradient distribution $\mathcal{G}(t)$ evolves non-negligibly during the time window used to compute moments, violating the *quasi-static assumption*.

**Mathematical Consequences:**

- The extremal set $I_{\max}(t)$ is time-dependent and may not converge.

- The diagonal limit in Theorem 4.7 becomes path-dependent: $\lim_{k,l \to \infty} \mathcal{C}(k, l)$ depends on the relative rates $k(t), l(t)$ versus the evolution of $\mathcal{G}(t)$.

- The coupling term $\mathcal{C}(k, l)$ may oscillate or drift, showing no stable asymptotic value.

**Observable Phenomena:**

- The overlap cardinality $m_\cap(t) = |J_{\max} \cap I_{\max}(t)|$ fluctuates between 0 and $\geq 1$.

- The $\mathcal{C}(k, k)$ curve is non-monotonic and shows transient spikes or dips (phase transition signals).

**Remedy and Physical Interpretation:** Non-stationarity occurs during critical learning periods (e.g., grokking onset, task switching in continual learning). This is not a failure of the theory but an opportunity:

1. *Time-scale separation*: Compute moments over intervals $\Delta t$ where $\mathcal{G}(t)$ is approximately constant.

2. *Moving averages*: Use $\mathcal{G}_{\mathrm{avg}}(t) = \frac{1}{\tau} \int_{t-\tau}^{t} \mathcal{G}(s) ds$ to filter high-frequency dynamics.

3. *Phase transition detection*: Violation of regularity conditions marks topological phase boundaries, providing a rigorous signal for phenomena like catastrophic forgetting.

### C.5 The Gradient Regularity as a Diagnostic Tool

Rather than viewing these conditions as restrictive assumptions, they serve as operational diagnostics:

- **Healthy Training:** Conditions (G1) and (G2) hold; gradient moments follow the predicted decomposition; coupling term $\mathcal{C}(k, l)$ is stable and bounded.

- **Critical Phase:** Condition (G1) violated (vanishing gap); $m_\cap(t)$ fluctuates; signals approach to ordered/disordered transition.

- **Pathological Landscape:** Condition (G2) violated (heavy tail); gradient moments diverge; indicates need for architectural or hyperparameter changes.

- **Dynamic Regime:** Time-dependence dominates; static moment analysis insufficient; signals need for time-resolved or averaged analysis.

## D  Numerical Implementation of High-Order Joint Moments

In Section 6.6, we deployed the microscopic lens at $p = 20.0$ to diagnose algorithmic crystallization. However, a naive computation of high-order moments $\sum |x_i|^p$ for large $p$ will inevitably cause floating-point overflow or underflow, resulting in `NaN` values during empirical tracking.

To strictly preserve the mathematical integrity of the generalized measurement spectrum $\mathcal{C}_\phi$ in finite-precision arithmetic, we apply the Log-Sum-Exp (LSE) trick. Let $u_i = \log |\theta_i|$ and $v_i = \log |g_i|$. The logarithmic joint moment of order $p$ is computed directly in the log-domain:

$$\log Z(p, p) = \log \left( \frac{1}{n} \sum_{i=1}^{n} |\theta_i|^p |g_i|^p \right)$$

$$= \log \sum_{i=1}^{n} \exp\left( p(u_i + v_i) \right) - \log n$$

Let $w_i = u_i + v_i$ and $w_{\max} = \max_i w_i$. We factor out the maximal element to strictly bound the exponential arguments:

$$\log Z(p, p) = p \cdot w_{\max} + \log \sum_{i=1}^{n} \exp\left( p(w_i - w_{\max}) \right) - \log n$$

Since $w_i - w_{\max} \leq 0$, the exponential term is strictly bounded within $(0, 1]$, completely eliminating the risk of overflow regardless of how large $p$ becomes. The marginal moments $\log M(p)$ and $\log G(p)$ are computed using the identical LSE stabilization. This ensures that the coupling term $\mathcal{C}(p, p)$ remains numerically exact even in the extreme thermodynamic limits.

# E  Geometric Origin of Disorder: Pathological Hessian Curvature

In Section 7, we hypothesized that the disordered phase ($m_\cap = 0$, leading to the strict negative divergence of $\mathcal{C}(k,k)$) is the macroscopic statistical manifestation of a microscopic geometric collapse. Here, we provide a formal mathematical translation of this phenomenon using the local Hessian matrix.

## E.1  Local Quadratic Approximation

Consider the local loss landscape $\mathcal{L}(\theta)$ around a local minimum or stable critical point $\theta^*$. The local geometry is governed by the Taylor expansion of the gradient:

$$g(\theta) = \nabla \mathcal{L}(\theta) \approx \mathcal{H}(\theta - \theta^*)$$

where $\mathcal{H} \in \mathbb{R}^{n \times n}$ is the Hessian matrix. To isolate the intrinsic geometric properties without the confounding effects of cross-parameter correlations, we assume a locally diagonalized eigen-basis where the parameters $\theta_i$ align with the principal curvature directions (a standard analytic assumption, e.g., LeCun et al., 1998). Furthermore, assuming zero-mean regularization (like weight decay) anchors the stable features near the origin, we can approximate the magnitude relationship as:

$$|g_i| \approx |\lambda_i||\theta_i|$$

where $\lambda_i$ is the local curvature (eigenvalue) associated with the parameter $\theta_i$.

## E.2  Mathematical Diagnosis of the Disordered Phase

By Definition, when the system is in the severe **disordered phase** (driving $\mathcal{C}(k,k) \to -\infty$), the parameter extreme set $J_{\max} = \arg\max_i |\theta_i|$ and the gradient extreme set $I_{\max} = \arg\max_i |g_i|$ are entirely disjoint ($J_{\max} \cap I_{\max} = \emptyset$).

Let $\theta_{\max} = \max_i |\theta_i|$ and $g_{\max} = \max_i |g_i|$. This disjoint topological condition strictly enforces two simultaneous geometric pathologies:

**1. Frozen Structural Capacity (Zero Curvature):** For any index $j \in J_{\max}$, the parameter is massive: $|\theta_j| = \theta_{\max}$. However, because $j \notin I_{\max}$ and the spectrum has a strict gap, its gradient is vanishingly small relative to the maximum: $|g_j| \ll g_{\max}$. Substituting this into our geometric equation yields:

$$|\lambda_j| \approx \frac{|g_j|}{|\theta_j|} = \frac{|g_j|}{\theta_{\max}} \longrightarrow 0$$

*Geometric interpretation:* The network's largest structural components (its dominant mathematical capacity) are stranded in perfectly flat, zero-curvature valleys. They act as "dead features" that absorb complexity penalties without providing optimization utility.

**2. Over-stressed Noise (Infinite Curvature):** For any index $i \in I_{\max}$, the gradient is maximal: $|g_i| = g_{\max}$. However, because $i \notin J_{\max}$, its parameter magnitude is microscopic relative to the dominant weights: $|\theta_i| \ll \theta_{\max}$ (often $|\theta_i| \to 0$ near initialization noise). Substituting this yields:

$$|\lambda_i| \approx \frac{|g_i|}{|\theta_i|} = \frac{g_{\max}}{|\theta_i|} \longrightarrow \infty$$

*Geometric interpretation:* The network's most intense optimization energy ($g_{\max}$) is desperately trying to update microscopic noise variables. For a tiny variable to generate a massive gradient, it must be traversing an infinitely sharp ravine in the loss landscape.

## E.3  Conclusion: The Eigenspectrum Inversion

Therefore, the mathematical divergence of the joint partition function $\mathcal{C}(k,k) \to -\infty$ is not merely a statistical anomaly; it rigorously diagnoses an **eigenspectrum inversion**. In a healthy network (the Ordered

Phase, $m_\cap \geq 1$), the maximal parameter and gradient align, yielding a stable, finite dominant curvature $|\lambda_{i^*}| \approx g_{\max}/\theta_{\max}$.

Conversely, the disordered phase exactly maps to a state where structural capacity corresponds to null-spaces ($\lambda \to 0$), while optimization stress is entirely borne by hyper-sensitive noise ($\lambda \to \infty$). This rigorously explains why models trapped in the disordered phase (e.g., early epochs of Grokking) fail to generalize: they are optimizing along pathologically sharp directions while their primary capacity remains unutilized.

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
