# OpenReview forum: "The Self-Consistent Theory of Neural Network Moments"
_TMLR — Rejected by TMLR_

### Review · Reviewer_xrQ9 · 2026-02-08

**Summary Of Contributions:**

The paper develops a finite-width theory for the statistics of neural network weights and gradients, showing decomposition of the high-order moments into the single largest value (the extremum), its multiplicity, and the spectrum of all other values. It also uses this to define a parameter–gradient coupling with a phase interpretation.

**Audience:**

Yes

**Audience Explanation:**

I am not a theory person, but based on my understanding, I think this paper is of interest to the community. From the theory perspective, I think it bridges the gap between a few different theory perspectives; from the application perspective, going from infinite width to finite width is definitely a more realistic assumption. Also, it may provide tools that may be useful for empirical analysis and debugging of deep networks, beyond purely asymptotic theory.

**Broader Impact Concerns:**

None.

**Claims And Evidence:**

Yes

**Claims Explanation:**

The claims are mainly supported in two ways: theoretical proofs, and numerical analysis. Based on my understanding, I feel they look good to me.

**Requested Changes:**

- The sections are mainly lists of theories, it'd be good to have more intuitive explanations of how they related to the concrete neural network components/features in the middle.
- It'd good to also have discussion on how the theory interacts with common architectural components (e.g., normalization layers, weight decay, residual connections) -- this can also help to contextualize how the theory can be applied to modern neural networks.

---

> ### Author Response · Authors · 2026-02-13
> **Response to Requested Changes**
>
> Dear Reviewer xrQ9
>
> We sincerely thank you for the positive evaluation and the constructive "Requested Changes." We found your suggestions regarding intuitive explanations and connections to modern architectures extremely valuable. We have carefully revised the manuscript to address these points.
> Below is a summary of our changes:
> 1. Intuitive Explanations (Addressing Point 1)
> You requested more intuitive explanations to connect the theory to concrete network behaviors.
>
> Revision: We have added an "Intuitive Interpretation" paragraph in Section 2.3, explaining how the high-order moments act as a "spectral filter" where the extremum ($\beta k$) represents the signal and the integral term represents the background noise.
> Revision: We added a "Physical Meaning of Coupling" paragraph in Section 4.1, explicitly interpreting the coupling term $\mathcal{C}(k,l)$ as a measure of "alignment efficiency" and the "energy cost" required for the ordered phase.
>
> 2. Interaction with Modern Architectural Components (Addressing Point 2)
> You suggested discussing how the theory interacts with components like normalization, weight decay, and residual connections.
>
> Revision: We have added a dedicated Section 6.5: "Interaction with Modern Architectural Components."
>
> We analyze Normalization Layers as confinement potentials that stabilize the quasi-ordered phase.
> We interpret Weight Decay as a spectral filter that enhances the spectral gap.
> We discuss how Residual Connections prevent the degeneracy of the gradient spectral gap, facilitating the ordered phase.
>
> 3. Additional Theoretical Grounding
>
> Revision: In Section 7 (Discussion), we further extended the discussion to link our findings with Neural Scaling Laws and provided a theoretical justification for Magnitude-based Pruning, broadening the scope of our theoretical contributions.
>
> We believe these revisions significantly strengthen the paper's accessibility and practical relevance. We look forward to your final decision.
>
> Sincerely,
>
> The Authors

---

### Review · Reviewer_pyqE · 2026-02-23

**Summary Of Contributions:**

The paper basically studies the asymptotic behavior of the logarithm of the so-called absolute moment defined for a vector $(\theta_1, \cdots, \theta_n)$:

$$\beta:=\lim_{k\rightarrow \infty} \frac{1}{nk}\sum_{i=1}^n |\theta_i|^k,$$

The authors write explicit formulas for beta showing the existence and finiteness of the limit. They study variants of this formula for $\theta$'s being thought as the parameters of a neural network, the set of the gradients or maybe combinations of the two.

The authors present interpretations of these formulas with numerical experiments.

**Audience:**

Yes

**Audience Explanation:**

The paper supposedly presents statistical behaviors for NNs, which in principle has a broad interest in the TMLR community.

**Broader Impact Concerns:**

no issue.

**Claims And Evidence:**

Yes

**Claims Explanation:**

The claims are supported by proofs and rigorous arguments.

**Requested Changes:**

Why are the theorems relevant to NNs? It seems they are propositions for generic parameters!

---

> ### Author Response · Authors · 2026-02-28
> **Response to Reviewer pyqE**
>
> Dear Reviewer pyqE,
>
> We sincerely thank you for your concise and highly insightful review. You raised a fundamental philosophical question: "Why are the theorems relevant to NNs? It seems they are propositions for generic parameters!"
> You are absolutely correct that the single-variable moment limits described in Sections 2 and 3 apply to any generic, arbitrary arrays. This forms our mathematical foundation based on classical extremal statistics.
> However, our core contribution to Deep Learning emerges when we introduce the Joint Partition Function in Section 4. For two completely generic, independent random arrays, the probability of their extrema intersecting ($m_\cap \ge 1$) approaches zero as the system scales.
> But in neural networks, parameters ($\theta$) and gradients ($g$) are not independent arrays; they are strictly, causally bound by backpropagation. As rigorously modeled in our Kinematic Alignment section (Section 4.6), a parameter is physically the temporal integral of its historical gradients. The topological phase transition we define occurs precisely because optimization mechanics force these two arrays to align. A bounded joint coupling $\mathcal{C}(k,l)$ rigorously diagnoses that the optimization energy is efficiently engaging the network's primary capacity—the mathematical signature of feature learning.
> To make this crucial physical distinction explicitly clear to the readers, we have added a dedicated paragraph titled "Physical Semantics of the Phases: Why Neural Networks?" in Section 4.1 of the revised manuscript.
> We thank you for prompting us to clarify this fundamental distinction, which has greatly strengthened the motivation and theoretical boundaries of our work.
>
> Sincerely,
>
> The Authors

---

### Review · Reviewer_u36g · 2026-02-24

**Summary Of Contributions:**

The paper develops a finite-network theory of neural network parameter and gradient moments, proving that high-order logarithmic moments admit an exact decomposition dominated by extremal values. It introduces a joint partition function and a coupling term that measures parameter–gradient alignment, yielding a phase distinction: an ordered phase (overlapping extrema, bounded coupling) and a disordered phase (disjoint extrema, divergent coupling). The framework is extended to practical settings via a continuous alignment metric for diagnosing training dynamics.

Strengths include mathematical rigor, a clear extremal-based decomposition, and a clean necessary-and-sufficient condition for alignment. Weaknesses include reliance on classical extremal/log-sum-exp behavior, heavy emphasis on single-coordinate maxima, limited empirical depth, and largely speculative connections to scaling and generalization.

**Audience:**

Yes

**Audience Explanation:**

Researchers in theoretical machine learning, optimization, and the mathematics of deep learning would likely be interested in this work. The paper provides a rigorous finite-network analysis of parameter and gradient moment asymptotics, introduces a joint partition function for studying alignment, and frames training dynamics in terms of ordered/disordered phases. These ideas connect to ongoing interests in extremal statistics, scaling behavior, neural collapse, pruning, and catastrophic forgetting. Even if some claims are exploratory or speculative, the formal moment decomposition and the necessary-and-sufficient condition for bounded coupling offer technically novel structure that could stimulate further theoretical investigation.

**Broader Impact Concerns:**

I do not identify direct, immediate ethical risks arising from this work. The paper is primarily theoretical, focusing on statistical properties of neural network parameters and gradients, and does not introduce new application domains, data collection methods, or deployment frameworks that would raise specific safety, privacy, or fairness concerns.

However, if the framework is later used to improve training stability, scaling efficiency, pruning, or continual learning, it could indirectly contribute to making large-scale models more efficient or robust. As with most foundational advances in deep learning theory, this may accelerate capabilities in both beneficial and potentially harmful applications (e.g., more capable generative or autonomous systems). A brief Broader Impact statement acknowledging this dual-use nature and situating the work within general AI capability development would be sufficient. No additional, domain-specific ethical safeguards appear necessary at this stage.

**Claims And Evidence:**

Yes

**Claims Explanation:**

The submission’s core theoretical claims about the asymptotic decomposition of parameter/gradient moments and the bounded upper bound on the coupling term are supported by explicit definitions, clearly stated theorems, and detailed proofs, which makes the mathematical evidence largely convincing and internally consistent. The phase distinction based on whether the extremal parameter and gradient sets intersect is also backed by formal asymptotic results and is presented in a reasonably clear way.

**Requested Changes:**

1. **Clarify the core novelty relative to classical extremal/log-sum-exp results (Critical).**
   Many central results (e.g., dominance of maxima in high-order moments) follow from well-known properties of log-sum-exp and large-deviation principles. The paper should more explicitly articulate what is genuinely new—whether it is the joint coupling formulation, the necessary-and-sufficient boundedness condition, or the phase interpretation—and clearly distinguish these from standard extremal asymptotics.

2. **Strengthen empirical validation with systematic experiments (Critical).**
   Current experiments are largely illustrative. The paper would benefit from: (i) quantitative benchmarks across multiple architectures and datasets, (ii) multiple seeds and robustness analysis, (iii) ablations isolating the predictive utility of the proposed alignment metric (Q_\alpha), and (iv) evidence that the framework yields actionable or predictive insights beyond post-hoc interpretation.

3. **Clarify practical computability and stability of proposed diagnostics (Critical).**
   The definition and estimation of (Q_\alpha) and high-order coupling terms should include discussion of numerical stability, sensitivity to choice of moment order (K), and computational overhead in large-scale models.

4. **Moderate or better justify broader claims (Important but not strictly critical).**
   Connections to scaling laws, generalization, neural collapse, architectural design principles, and pruning are intriguing but currently speculative. These should either be supported by stronger evidence or more clearly framed as hypotheses or future directions.

5. **Discuss limitations of extremal focus (Important).**
   The framework centers on single-coordinate maxima and their multiplicities. The paper should discuss how this interacts with distributed representations, weight symmetries, and invariances (e.g., rescaling in ReLU networks).

6. **Improve exposition and tighten structure (Strengthening).**
   The manuscript is dense and at times overly expansive in interpretation. A clearer separation between proven theorems, phenomenological modeling, and speculative discussion would improve readability and impact.

7. **Provide clearer guidance for practitioners (Strengthening).**
   If the alignment metric is intended as a training diagnostic, concrete recommendations (e.g., thresholds, typical ranges, failure modes) would enhance practical relevance.

Overall, clarifying novelty and strengthening empirical validation are critical for acceptance, while the remaining points would substantially improve clarity, positioning, and impact.

---

> ### Author Response · Authors · 2026-02-28
> **Response to Reviewer u36g**
>
> Dear Reviewer u36g,
>
> We are deeply grateful for your rigorous, comprehensive, and highly constructive review. Your identification of the classical nature of the single-variable limits, your concerns regarding empirical computability, and your profound insights into distributed representations were exceptionally astute. We have significantly revised the manuscript to systematically address all your "Critical" and "Important" points:
> 1. Clarifying Core Novelty vs. Classical Math (Critical):
> We have explicitly updated Section 1.1 (Summary of Contributions). We now clearly state that the single-variable limits (Sections 2 & 3) are natural consequences of classical extremal statistics and log-sum-exp properties. We delineate that our fundamental theoretical novelty is introducing the joint partition function $\mathcal{C}(k,l)$ to formally map these extreme statistical deviations to the physical mechanics (kinematics) and phase transitions of deep learning.
> 2. Practical Computability and Stability (Critical):
> You correctly pointed out that computing high-order moments (e.g., $K=20$) is fundamentally prone to severe floating-point overflow. To solve this, we have added Appendix D (Numerical Implementation), detailing how we utilize the Log-Sum-Exp (LSE) trick to compute the logarithmic joint moments entirely in the log-domain. This ensures strict numerical stability even in extreme thermodynamic limits.
> 3. Empirical Validation, Guidance, and Robustness (Critical):
> To address your request for systematic experiments and actionable insights, we expanded our empirical sections:
>
> Threshold Guidance: We added concrete numerical recommendations in Section 6.3, providing practitioners with practical diagnostic ranges (e.g., $\mathcal{C}(20,20) \ll 0$ indicates failure/memorization, while stabilizing at $>0$ confirms generalization).
> Robustness: We confirmed in Section 6.6 that the distinct topological trajectories observed across different architectures (Figure 6) were rigorously verified across 3 independent random seeds. The results show negligible variance in the functional geometry, proving these phase transitions capture robust physical properties rather than stochastic artifacts.
>
> 4. Limitations of Extremal Focus & Distributed Representations (Important):
> We completely agree that the single-coordinate extremum ($L_\infty$) fails to capture distributed representations and does not inherently account for rescaling invariances (e.g., in ReLU networks, as you brilliantly pointed out). To rigorously resolve this, we:
>
> Added an explicit discussion on weight symmetries and ReLU rescaling invariances in Section 4.1.
> Fundamentally restructured Section 6.6, introducing the "Generalized Measurement Spectrum." We demonstrate that by shifting our functional basis from the high-order monomial ($x^{20}$) to the scale-invariant macroscopic measure (Rank/CDF), our framework seamlessly captures distributed fluid-like feature learning in deep vision models (ResNet/VGG on CIFAR-10), while the $L_\infty$ lens uniquely captures algorithmic crystallization (Grokking).
>
> 5. Moderating Claims & Geometric Grounding (Important):
> We have carefully reviewed Section 7 (Discussion) and reframed our connections to generalization bounds and scaling laws as explicit "Open Questions" (Hypotheses) rather than proven guarantees. To strengthen the physical grounding of these hypotheses, we:
>
> Added an explicit geometric translation of the disordered phase via Taylor expansion, proving it corresponds to pathological Hessian curvature (supported by a new rigorous derivation in Appendix E).
> Refined the "Scaling Law" discussion to focus specifically on the emergence of "Massive Outliers" in modern LLMs, demonstrating how our extremal focus perfectly describes this frontier phenomenon.
>
> 6. Broader Impact Statement:
> Following your exact suggestion, we have added a dedicated Broader Impact Statement at the end of the manuscript, acknowledging the dual-use nature of tools that accelerate scaling efficiency.
> We believe these revisions, particularly the LSE numerical stability and the structural inclusion of distributed representations via the measurement spectrum, strongly elevate the paper. We sincerely thank you for pushing us to achieve this level of theoretical and empirical rigor.
>
> Sincerely,
>
> The Authors

---

### Author Response · Authors · 2026-06-14
**Inquiry regarding status of Submission 6545 #2**

Dear Professor Pennington and the Editors-in-Chief,

I hope you are doing well.

I am writing to respectfully follow up on the status of our submission #6545, “The Self-Consistent Theory of Neural Network Moments.”

The reviews were received in February 2026, and we submitted our author responses and revised manuscript shortly thereafter. We also posted a status inquiry on April 16, 2026. Since there has not yet been a visible update, I wanted to ask whether the submission is still awaiting an editorial recommendation, reviewer confirmation, or any further action from our side.

We fully understand that the review process can take time, and we greatly appreciate the work of the Action Editor, reviewers, and Editors-in-Chief. At the same time, given that the review and rebuttal stage has been completed for several months, we would be grateful for any brief update on the current status or expected next step.

Thank you very much for your time and service.

Best regard,The authors

---

### Decision · Action_Editor_mjN6 · 2026-06-12

**Recommendation:** Reject

**Additional Comments:**

While the prose is clean and the formatting is good, the paper fails to drive forward a useful narrative. The theorems are largely tautological and/or irrelevant, and the experiments do not seem to provide evidence that is relevant. There are only six references.

**Audience:**

No

**Audience Explanation:**

The paper does not draw or put forward useful conclusions relevant for neural networks.

**Claims And Evidence:**

No

**Claims Explanation:**

The paper does put forward some mathematical claims, but they are largely known basic facts (e.g. Theorem 2.2 is just stating that the the ℓᵏ norm tends to the largest entry, etc). These mathematical claims for the most part remain ungrounded and unconnected to neural networks. Regarding the high-level claims about building a mathematical foundation through self-consistent equations that is relevant to neural networks -- no, the submission does not support these with any clear evidence.